



# Towards monitoring CO₂ source-sink distribution over India via inverse modelling: Quantifying the fine-scale spatiotemporal variability of atmospheric CO₂ mole fraction

Vishnu Thilakan[1,4], Dhanyalekshmi Pillai[1,4], Christoph Gerbig[2], Michal Galkowski[2,3], Aparnna Ravi[1,4], and Thara Anna Mathew[1]

[1]Indian Institute of Science Education and Research Bhopal (IISERB), Bhopal, India

[2]Max Planck Institute for Biogeochemistry, Jena, Germany

[3]AGH University of Science and Technology, Kraków, Poland

[4]Max Planck Partner Group (IISERB), Max Planck Society, Munich, Germany

*Correspondence to*: Dhanyalekshmi Pillai (dhanya@iiserb.ac.in, kdhanya@bgc-jena.mpg.de)

**Abstract**

The prospect of improving the estimates of CO₂ sources and sinks over India through inverse methods calls for a
comprehensive atmospheric monitoring system involving atmospheric transport models that make a realistic accounting of atmospheric CO₂ variability. In the context of expanding atmospheric CO₂ measurement networks over India, this study aims to investigate the importance of a high-resolution modelling framework to utilize these observations and to quantify the uncertainty due to the misrepresentation of fine-scale variability of CO₂ in the employed model. The spatial variability of atmospheric CO₂ is represented by implementing WRF-Chem at
a spatial resolution of 10 km × 10 km. We utilize these high-resolution simulations for sub-grid variability calculation within the coarse model grid at a horizontal resolution of one degree (about 100 km). We show that the unresolved variability in the coarse model reaches up to a value of 10 ppm at the surface, which is considerably larger than the sampling errors, even comparable to the magnitude of mixing ratio enhancements in source regions. We find a significant impact of monsoon circulation in sub-grid variability, causing ~3 ppm
average representation error between 12 – 14 km altitude ranges in response to the tropical easterly jet. The cyclonic storm Ockhi during November 2017 generates completely different characteristics in sub-grid variability than the rest of the period, whose influence increases the average representation error by ~1 ppm at the surface. By employing a first-order inverse modelling scheme using pseudo observations from nine tall tower sites over India and a constellation of satellite instruments, we show that the Net Ecosystem Exchange
(NEE) flux uncertainty solely due to unresolved variability is in the range of 6.3 to 16.2 % of the total NEE. We illustrate an example to test the efficiency of a simple parameterization scheme during non-monsoon periods to capture the unresolved variability in the coarse models, which reduces the bias in flux estimates from 9.4% to 2.2 %. By estimating the fine-scale variability and its impact during different seasons, we emphasise the need for implementing a high-resolution modelling framework over the Indian subcontinent to better understand
processes regulating CO₂ sources and sinks.



## 1 Introduction

Accurate assessment of sources and sinks of $CO_2$ is essential in planning and implementing the mitigation
strategies for greenhouse gas emission and associated climate change. However, estimations of $CO_2$ fluxes
contain significant uncertainties which increase even more with finer spatial scales such as those required for the
climate change mitigation policies at regional and national levels (e.g., Ciais et al., 2014; Li et al., 2016;
Cervarich et al., 2016). By using atmospheric $CO_2$ concentration measurements, the $CO_2$ fluxes can be estimated
by a multi-constrained observation-modelling approach, often referred to as top-down approach or inverse
modelling (Enting, 2002). For about two decades, these top-down approaches are being widely used to
understand the modifications in the carbon cycle through natural and anthropogenic induced environmental
changes (Bousquet, 2000; Schimel et al., 2001; Rödenbeck et al., 2003; Patra et al., 2005). In addition to the
observations, the inverse modelling system makes use of an atmospheric transport model (forward model),
which determines the distribution of $CO_2$ concentration. Thereby, the inverse optimization approach derives the
surface fluxes that are consistent with measured concentration. The UNFCCC (United Nations Framework
Convention on Climate Change) has acknowledged the increasing capability of inverse modelling to
systematically monitor GHG concentrations (Bergamaschi et al., 2018).

Most of the inverse modelling systems relies on global atmospheric transport models with coarse horizontal
resolution (often greater than one degree) (Rödenbeck et al., 2003; Peters et al., 2007; Rödenbeck et al., 2018a,
b; Inness et al., 2019). However, regional estimation of fluxes using global models is hindered by the inability of
transport models to represent observed $CO_2$ variability. The observed variability, as seen from the spatial and
temporal distribution of atmospheric $CO_2$, is highly correlated with the space and time scales of weather systems
(Parazoo et al., 2011). This explains the presence of large model-data mismatches in regions where mesoscale
circulation is predominant (Ahmadov et al., 2007). Wind speed, wind direction, and height of the planetary
boundary layer (PBL) are the critical variables that determine the atmospheric $CO_2$ variability. Strong wind
normalizes other small-scale variations in observed concentration due to mixing, and the predictability can be
higher during these conditions (Sarrat et al., 2007). The height of PBL is an essential variable in modelling $CO_2$
concentration since the atmospheric $CO_2$ is subjected to rapid mixing up to this altitude. Hence, for a given
location with a negative gradient in $CO_2$ vertical distribution, the overestimation of PBL height leads to
underestimation of $CO_2$ concentration and vice-versa (Gerbig et al., 2008).

Another important variable that impacts the $CO_2$ variability is the heterogeneous topography. When the small-
scale orographic details are not adequately represented in the models, they can lead to representation errors in
$CO_2$ simulations as large as 3 ppm at scales of 100 km (Tolk et al., 2008; Pillai et al., 2010). The horizontal
gradients in $CO_2$ concentrations can go up to values of 30 ppm within a spatial scale of 200 km, depending on
the land surface heterogeneity (van der Molen and Dolman, 2007). Previous studies based on airborne
measurements reported that transport models need a spatial resolution which is smaller than 30 km to be able to
represent $CO_2$ spatial variability in the continental boundary layer (Gerbig et al., 2003). Significant efforts have
been invested to derive fluxes by taking to account these fine-scale variations (e.g., Gerbig et al., 2003; Lauvaux
et al., 2009; Carouge et al., 2010; Pillai et al., 2011, 2012; Broquet et al., 2013) over North American and
Eurasian domains in the past decade. However, there still exists lower confidence in estimates over the region,



where there is a lack of both advanced modelling systems at relevant spatio-temporal resolutions and good coverage of ground-based monitoring stations.

In the context of the Indian sub-continent, the inverse-based estimation of fluxes at fine scales is essentially new; hence many questions remain. A number of monitoring sites measuring atmospheric greenhouse gases

have become available in India from the last decade (Tiwari et al., 2011; Lin et al., 2015). Aside from the ongoing progress in augmenting observational data streams, it remains challenging to assimilate these data for deducing process-specific information effectively (e.g., McKain et al., 2012; Bréon et al., 2015; Pillai et al., 2016). The limitation of these coarse models on representing observations over the Indian subcontinent is reflected in the analysis made by Patra et al. (2011).

The seasonally reversing South Asian monsoon system is a prominent meteorological phenomenon affecting the Indian subcontinent, which is also expected to influence the terrestrial-atmosphere flux exchanges. Various studies have demonstrated the role of Indian monsoon circulations on regional atmospheric transport by strong south westerlies during the summer monsoon (June to September) and by north easterlies during the winter monsoon (October to November) (e.g., Goswami and Xavier, 2005; Krishnamurthy and Shukla, 2007). The

monsoon convection that transports the boundary layer air into the free atmosphere (mainly to the upper troposphere and to the lower stratosphere with the help of diabatic heating (Vogel et al., 2019)) complicates atmospheric transport simulations (Willetts et al., 2016). Moreover, a significant component of flux variations can arise from biospheric fluxes (Schimel et al., 2014), which is influenced by variables such as rainfall, availability of radiation, and temperature (Chen et al., 2019). Several studies showed that the monsoon system

substantially impacts vegetation growth, generating distinct spatio-temporal patterns of the biogenic fluxes (e.g., Gadgil, 2003; Valsala and Maksyutov, 2013). It is noteworthy that the cropping patterns over India have a strong dependence on seasons and are mainly determined by dry and wet seasons for nearly 65 to 70 % of the country's area except over north-eastern and south-western (Western Ghats) regions of India. Therefore, employing a higher resolution modelling over Indian subcontinent is desirable, which can take into account the

fine-scale variations generated by both mesoscale transport processes and surface flux patterns.

The study focuses on accounting the unresolved sub-grid scale variability when employing current generation global models. Assimilation of observations in an inverse framework requires the characterization of these error structures at relevant scales that can be utilized to retrieve source-sink distribution over India. The main objectives of this paper are to describe and quantify the expected spatiotemporal variability of atmospheric $CO_2$

that is not resolved by the current generation global models, quantify to what extent these variations cause uncertainty in flux estimations, and assess how these uncertainties can be minimized by modelling the sub-grid variations in the global models. Specifically, we address the following questions: 1) how good is the level of agreement among global transport models that are used in the current generation inversion systems for predicting atmospheric $CO_2$ concentrations over Indian subcontinent? 2) how large are the variations of

atmospheric $CO_2$ that are unresolved by global and regional models which operate at a spatial scale of 1°× 1° and 0.5°× 0.5° respectively? 3) what is the role of seasonal changes and synoptic events on generating different patterns in these sub-grid variations of $CO_2$? 4) how much is the uncertainty in the inverse-based flux estimation, caused by these unresolved variations in the coarse models when utilizing a given network of surface observations and satellite measurements over the domain? 5) how effectively can we capture the key aspects of



the variability and take into account for it in flux estimations? We present an analysis based on our high-resolution simulations at a spatial resolution of 10 km × 10 km, and the sub-grid variability derived using these high-resolution simulations. By designing a pseudo surface observation network and hypothetical satellite measurements over the domain, we investigate the impact of these unresolved variations on the regional flux estimations and assess how a simple parameterization scheme can help in reducing these errors in the global model.

In this article, we present results based on the analyses of the high-resolution simulations for the months of July and November 2017. The year 2017 is characterized by neutral Indian Ocean Dipole conditions over the Indian Ocean and with the beginning of a mild La Nina over the Pacific by the end of the year. The month of July represents a monsoon period when the biospheric activity is significant together with atmospheric convection activities. On the other hand, the month of November is more representative of post-monsoon wintertime in Indian subcontinent. At the same time, the period from 23 to 30 November 2017 provides us with the opportunity to study the impact of a synoptic event on contributing to $CO_2$ variability. To our knowledge, there is no comprehensive published study of this kind over the Indian subcontinent until now assessing the magnitude and impact of temporal and spatial variability exhibited by atmospheric $CO_2$.

The outline of the paper is as follows. Section 2 describes our modelling system, data and approaches including the methods used for estimating the sub-grid scale variability of $CO_2$. Inter-model comparisons using global models and our investigations on spatial variability are discussed in section 3. Through the global model comparisons and spatial variability analysis, we highlight potential difficulties for estimating current $CO_2$ budget assessments over India and provide the quantification of expected spatiotemporal variability as well as its impacts. Finally, we provide the implications of our findings in section 4, suggesting the ways forward to yielding an improved estimation of $CO_2$ budgets over India.

## 2 Data & Methodology

A major part of this study is performed using the simulations generated by our high-resolution modelling system (see Sect. 2.1). For providing a more comprehensive overview of the inter-model agreements among the existing model simulations over Indian subcontinent, we use a variety of global model simulations as explained in Sect. 2.2. These global model outputs are derived from the inverse model simulations, which estimates the source-sink distributions of $CO_2$ and then three-dimensional $CO_2$ concentration fields are generated consistent with the optimized posterior fluxes. These models differ in terms of the model configuration, observational datasets that were assimilated (e.g., data from surface monitoring stations, aircraft missions, ship cruises, AirCore balloon soundings, and satellite's total column retrievals), prior datasets, and spatiotemporal resolutions. For quantifying the spatial variability of $CO_2$ over India, we use our high-resolution simulations, focusing on the representation error approach as explained in Sect. 2.3. Following a first-order assumption, a simple method is used to estimate the impact of the derived sub-grid scale variations on flux estimations over India (see Sect. 2.4).

### 2.1 WRF-Chem GHG Modelling System

For this study, we use the modelling system WRF-Chem GHG in which the Weather Research and Forecasting model (WRF) version 3.9.1.1 (Skamarock et al., 2008) is coupled with the greenhouse gas module (WRF-



Chem-GHG, Beck et al., 2011), implemented as part of the WRF-Chem distribution (WRF-Chem, Grell et al., 2005). For simulating the atmospheric transport, the model uses fully compressible Eulerian non-hydrostatic equations on Arakawa C- staggered grid, conserving mass, momentum, entropy and scalars (Skamarock et al., 155 2008). In the WRF-Chem GHG (hereafter referred simply as WRF-GHG), we use the passive tracer chemistry option to simulate changes in $CO_2$ mixing ratios associated with surface fluxes and atmospheric transport. We utilize a biospheric model and emission inventory data to simulate atmospheric $CO_2$ enhancements associated with biogenic and emission fluxes as described in Sect. 2.1.1 and 2.1.2. Table. 1 summarizes the model configuration, including physics parameterizations and input data used in this study.

The model domain covers a region spanning from 65°E to 100°E and 5°N to 40°N, configured in a Lambert conformal conic (LCC) projection with 307 × 407 grid points. The spatial resolution of the grid is 10 km × 10 km and model time-step of 60 s. We have used model output with a temporal resolution of 1 hour for this study. The simulations are performed using 39 vertical levels with the model top at 50 hPa and 10 levels within the lowest 2 km. WRF-GHG simulations are performed for the entire July and November 2017 for this 165 study. Implementation of the WRF-GHG system over Indian subcontinent enables us to customize it according to the domain features and build a state-of-the-art modelling system, which eventually estimates $CO_2$ fluxes through regional inverse systems. The potential of the WRF-GHG model in simulating fine-scale spatial variability is also established in previous studies (Ahmadov et al., 2009; Pillai et al., 2011; Park et al., 2018).

### 2.1.1 Representation of biospheric fluxes

We use the Vegetation Photosynthesis and Respiration Model (VPRM) in the modelling system to calculate Net Ecosystem Exchange (NEE) representing the biospheric fluxes (Mahadevan et al., 2008). VPRM is a diagnostic biosphere model, which utilizes remote sensing products: Enhanced Vegetation Index (EVI) and Land Surface Water Index (LSWI) derived from reflectance data of the Moderate resolution Imaging Spectroradiometer (MODIS) as well as meteorological data: solar radiation and air temperature. In this study, these hourly NEE 175 calculations are performed within WRF-GHG, simultaneously with the meteorology simulations in which NEE is calculated as a sum of gross ecosystem exchange (GEE) and ecosystem respiration ($R_{eco}$). VPRM, in this case, uses the meteorological data provided by WRF-GHG. VPRM uses the SYNMAP vegetation classification (Jung et al., 2006) as well as EVI and LSWI from MODIS surface reflectance data at a resolution of 1 km and 8 days. We aggregate these indices specific for different vegetation types onto the LCC projection for the entire domain 180 at the model's spatial resolution. A number of studies have used VPRM for other regions around the world in which derived NEE shows good prediction skills for hourly to monthly timescales (Ahmadov et al., 2009; Pillai et al., 2011; Liu et al., 2018; Park et al., 2018).

### 2.1.2 Representation of emission fluxes

Anthropogenic $CO_2$ emission fluxes are prescribed from the Emission Database for Global Atmospheric 185 Research (EDGAR) dataset, version 4.3.2, provided at a horizontal resolution of 0.1° × 0.1°. We disaggregate the available annual emission data into hourly emissions, using the country and sector-specific temporal profiles of $CO_2$ emissions available at the product website (https://edgar.jrc.ec.europa.eu) (Steinbach et al., 2011; Kretschmer et al., 2014). To represent biomass burning emission, we have used data from the Global Fire Assimilation System (GFAS) with a spatial resolution of 0.1° × 0.1° and a temporal resolution of one day.





GFAS is based on satellite data, which calculate the fire emission by assimilating fire radiative power (FRP) observations from MODIS instruments (Kaiser et al., 2012). All these flux data are gridded and projected to WRF-GHG's model domain.

### 2.1.3 Initial and boundary conditions

Meteorological and chemical initial and boundary conditions are required in WRF-GHG to account for initial
state and inflow or background flow. The initial and lateral boundary conditions for the meteorological variables, including horizontal wind components, pressure, specific humidity, sea surface temperature (SST), and the necessary surface initialization fields are obtained from the ERA5 dataset, the European Centre for Medium-Range Weather Forecasts (ECMWF), extracted at a horizontal resolution of 25 km and a temporal resolution of 1 hour (Hersbach et al., 2020). In the case of $CO_2$ tracers, the initial and lateral boundary
conditions are obtained from the Copernicus Atmosphere Monitoring Service (CAMS, 2.2.4). (Massart et al., 2016; Agusti-Panareda et al., 2019). We have used the dry air mole fractions of $CO_2$ from the CAMS-GHG, which has a temporal resolution of 6 hour and horizontal resolution of $0.5° \times 0.5°$ with 137 vertical levels. The data is taken from the greenhouse gas analysis experiment, which is conducted with the objective of providing realistic 3D fields of atmospheric GHG concentrations in dry air mole fractions. Note that there exists a CAMS
product at 9 km × 9 km resolution, which is in the developmental phase and not yet available to the general public (personal contact: Anna.Agusti-Panareda@ecmwf.int).

We have utilized a different simulation strategy to take advantage of assimilated meteorological fields from ECMWF. The model is reinitialized each day with ECMWF assimilated data at 00.00 UTC after a spin-up period of 6 hours started from 18.00 UTC of the previous day. In the case of $CO_2$, the initial values at the
beginning of the restart hour are taken from the last hour (Ahmadov et al., 2012).

### 2.2 Global atmospheric transport models

In addition to our high-resolution simulations, we have used other optimized products at global scales and examined the representation of $CO_2$ variability over the Indian subcontinent. Four global inverse modelling products - CarbonTracker, CarboScope, LSCE v18r3 and LSCE FT18r1- available during the year 2017 are
used for our analysis. These global models depend on different formulations (e.g., transport and the employed inversion methodology) and associated data (e.g., the choice of measured atmospheric mixing-ratio observations and emission fluxes) in simulating the underlying processes. We have used the daytime (11:30 to 16:30 local time) concentration values from all these models for the analysis. A brief description of each model is followed (see Table. 2 for more details).

### 2.2.1 CarbonTracker (CT-2019B)

CarbonTracker is a global reanalysis system for $CO_2$ fluxes developed by the Earth System Research Laboratory (ESRL) at the National Oceanographic and Atmospheric Administration (NOAA) (Peters et al., 2007; Jacobson et al., 2020). We have used $CO_2$ mole fractions from the CarbonTracker (CT2019B version), having a horizontal resolution of $3° \times 2°$ with 25 vertical levels and a temporal resolution of 3 hours.

### 2.2.2 CarboScope


In this study, we have used the atmospheric $CO_2$ concentration from the CarboScope Inversion v2020 with a horizontal resolution of $5° \times 3.8°$ with 19 vertical levels and a temporal resolution of 6 hours (Rödenbeck et al., 2003; Rödenbeck et al., 2018a, b). The major focus of this inversion system is the inter-annual variability of the $CO_2$ fluxes, homogeneously covering time periods containing all data records to avoid spurious jumps.

**2.2.3 LSCE**

This atmospheric inversion system (Chevallier et al., 2005; Chevallier et al., 2010; Chevallier, 2013) provides atmospheric $CO_2$ concentration with horizontal resolution $3.7° \times 1.8°$ and 39 vertical levels and temporal resolution of 3 hours. Version FT18r1 (hereafter LSCE FT in this manuscript), which we utilized for the current study uses satellite retrievals from the Orbiting Carbon Observatory (OCO-2) for optimization of $CO_2$ fluxes.
We also used version v18r3 (hereafter LSCE in this manuscript), which utilizes surface observations for the optimization.

**2.3 Quantification of spatial variability**

For quantifying the spatial variability due to sub-grid scale processes which cannot be resolved by the coarse resolution models, we follow the approach as described in Pillai et al. (2010). The term 'representation error'
indicates the mismatch between the scales of model simulations and observations collected (Pillai et al., 2010; Janjić et al., 2017). In other words, the representation errors arise due to unresolved scales, which could not be captured by the model. Here we calculate the representation errors in the coarse resolution models, which can be fully resolved by implementing the high-resolution model. For estimating the representation error in a coarse model with a typical spatial resolution of $1° \times 1°$, we have calculated the standard deviation of $CO_2$ dry air mole
fraction simulated by the WRF-GHG model within the coarse grid boxes of $1° \times 1°$ as follows:

$$\sigma_{CO_2} = \sqrt{\frac{1}{n-1}\sum_{j=1}^{n}(m_j - \overline{m})^2} \qquad (1)$$

where $\overline{m} = \frac{1}{n}\sum_{j=1}^{n} m_j$

$n$ is the number of $0.1°$ boxes inside the coarser grid cell of $1° \times 1°$; $m$ is the $CO_2$ dry air mole fraction corresponding to $0.1°$ boxes; and $\overline{m}$ is the average within the coarser grid cell. So, the estimated values represent
the sub-grid scale variability within the coarse model grid cell which is having a horizontal resolution of $1° \times 1°$. As the observations can also be made by the space-borne instruments, we extend the analysis to column-averaged dry air mole fraction ($XCO_2$) as measured by the satellite instrument. i.e., $m$ represents either $CO_2$ at the second model level or $XCO_2$.

The surface representation errors are calculated using the model simulations from the second model level
(~200m) to avoid the inconsistency that can be generated from inputting emission fluxes at the first model level. Representation errors are calculated separately for daytime (11:30 to 16:30 local time) and nighttime (23:30 to 4:30 local time) to account for the difference in the sub-grid scale process during these times. To disentangle the correlated term from the total representation error, the monthly averages of representation errors are taken. The effect of random errors can be minimized when averaging over long time periods. However, we have also
analysed the sub-monthly temporal variations in the representation errors by using representation errors



calculated at an hourly time scale (Sect. 3.2.2). Note that all discussions presented in this study except for Sect. 3.2.2 are based on monthly averaged values of representation errors. Both July and November are used to understand the differences in the variability during summer and winter.

Due to the paucity of adequate ground-level observations over India, satellite observations play an essential role
in the estimation of $CO_2$ fluxes. Satellite observations can provide column average $CO_2$ ($XCO_2$) concentration with a precision of 1 to 1.5 ppm (O'Dell et al., 2012; Wunch et al., 2017). In order to utilize these satellite observations, the transport models being used in the inverse estimation must be highly accurate as small errors in atmospheric transport simulations (a few tenths of ppm) can cause significant biases (a few hundreds of MtC/year) in the inferred surface carbon budget at the scale of subcontinents (Chevallier et al., 2010). To assess
the ability of current-generation models in utilizing the $XCO_2$ over India, we have also performed a similar representation error analysis for $XCO_2$.

**2.4 Estimation of representation error induced flux uncertainty using pseudo measurements**

**2.4.1 Using surface measurements**

In order to quantify the impact of representation errors on flux estimations when utilizing surface measurements,
we have devised the following strategy. For this, we use nine $CO_2$ surface monitoring sites representing various geographical regions over India (Fig. 1). Not all these observation stations are currently fully operational or having continuous measurements. We have performed an observation system simulation experiment (OSSE) using high-resolution $CO_2$ simulations generated by the WRF-GHG model for each of these stations. We focus on the biospheric flux component, NEE. For deducing the contribution of the representation error to the
biospheric flux uncertainty, we have taken the following assumptions: 1) there are no model/observation errors other than representation error, and 2) the model captures the spatial and temporal patterns of flux correctly. This means that the difference between observations (OSSE) and simulations of atmospheric $CO_2$ is the representation error in ppm. Additionally, as a first-order simplification for the inversion, we assume that the footprints of each observation site span for a radius of 200 km based on our analysis using the Stochastic Time-
Inverted Lagrangian Model (STILT, Lin et al., 2003) and approximation of area covering of 50-90 percentile of contributions from STILT-derived footprints of each station. The STILT is driven with ECMWF IFS (Integrated Forecasting System) meteorological fields and the trajectories are calculated based on 100 virtual particles that are released for each time interval and location. The residence time of particles in the surface layer is weighted by the atmospheric density to derive the footprints of each location.

Through our simple least-square inversion approach, we retrieve monthly NEE by utilizing hourly $CO_2$ observations (from OSSE) and simulations over a month. Both the observation and simulation vector have 6480 (=1×9×30×24) elements for a month having 30 days, and the state vector has 9 (=1×9) elements corresponding to scaling factors of monthly fluxes over regions around 9 sites. In other words, each site has been assigned with one scaling factor for NEE, and there is a total of 9 scaling factors for a given month. As it is the case for OSSE,
we know the flux $\Phi_{true}$ ("true" flux) and can be utilized for calculating the deviation of posterior fluxes from the true fluxes. Here we use the VPRM NEE fluxes as the "true" fluxes in OSSE (see Sect. 2.1.2). We use the unit vector $\lambda$ as initial scaling factors. The linear inversion provides the optimized scaling factors $\lambda_{retr}$ and these scaling factors are applied to $\Phi_{true}$ within the area covering of 50-90 percentile of the footprints in order to



derive the posterior fluxes $\mathbf{\Phi}_{retr}$. For the rest of the region outside the footprint area of each site, the $\mathbf{\Phi}_{retr}$
remains the same as $\mathbf{\Phi}_{true}$.

The optimized monthly scaling factors ($\boldsymbol{\lambda}_{retr}$) of fluxes can be obtained by minimizing the following cost
function $J(\boldsymbol{\lambda})$:

$$J(\boldsymbol{\lambda}) = \frac{1}{N}\sum_{n=1}^{N} (\boldsymbol{y}_{OSSE} - \boldsymbol{y}_{sim})^2 \qquad (2)$$

where N is the size of observation and simulation vectors, $\boldsymbol{y}_{OSSE}$ is given by:

$$\boldsymbol{y}_{OSSE} = \mathbf{H} . \boldsymbol{f}_m(\boldsymbol{\lambda}) \qquad (3)$$

where $\mathbf{H}$ is the transport operator and $\boldsymbol{f}_m(\boldsymbol{\lambda})$ is the flux model in which a subset of parameters $\boldsymbol{\lambda}$ out of total
model parameters $\boldsymbol{p}$ will be optimized. Here $\boldsymbol{f}_m(\boldsymbol{\lambda})$ is taken as linearly dependent on $\boldsymbol{\lambda}$ ; hence can be expressed
as

$$\boldsymbol{f}_m(\boldsymbol{\lambda}) = \mathbf{\Phi}_{true} . \boldsymbol{\lambda} \qquad (4)$$

Note that $\boldsymbol{y}$ represents the biospheric contribution (in ppm) of the atmospheric $CO_2$ mixing ratio. By minimizing
the cost function as given in Eq. (2), we get the optimized estimate of scaling factors $\boldsymbol{\lambda}_{retr}$.  By this inverse
design, the deviation of posterior fluxes from the true fluxes is thus the uncertainty in retrieved fluxes, $\mathbf{S}_{rep}$ , that
arises solely due to the contribution from the representation error and is calculated as:

$$\mathbf{S}_{rep} = | \mathbf{\Phi}_{true} - \mathbf{\Phi}_{retr} | = | \mathbf{\Phi}_{true} - \mathbf{\Phi}_{true} . \boldsymbol{\lambda}_{retr} | = |(1 - \boldsymbol{\lambda}_{retr}) \mathbf{\Phi}_{true} | \qquad (5)$$

Note that by following the above inversion design and assumptions, there is a high likelihood of
underestimating the impact of the modelling error on flux estimations since we have not considered other
sources of uncertainties such as model transport uncertainty and inappropriate prior assumptions. Thus, the
quantification of flux uncertainty using this approach can be inferred as the lower bound of the uncertainty (i.e.,
the minimum flux uncertainty one may expect while estimating fluxes using a model with a grid cell of 1°× 1°
and 9 stations with representativeness of 200 km).

### 2.4.2 Using satellite measurements

Following the approach as given in Sect. 2.4.1, we have estimated the NEE flux uncertainty when using satellite
observations over the model domain. For this, we have assumed an idealized dense spatial sampling of a
constellation of satellites which sample as frequently as possible over each location with a wide swath enabling
to cover a large geographical area, typically a few hundred kilometres. As our focus is to deduce the flux
uncertainty solely due to the representation error, we assumed no measurement errors. In practice, the satellite
spectra are altered by atmospheric scattering by air molecules and the presence of aerosols and clouds, which
contribute to the measurement error. The presence of clouds significantly influences the number of available
(quality-filtered) observations over a region. To take this into account, only cloud-free observations are
considered. For this, we use cloud fraction derived from the radiance and reflectance measurements by MODIS
aboard NASA's Terra satellite product (MOD35).



Based on WRF-GHG simulations, we perform the analysis for November at 10.30 AM each day and sample the data after considering the cloud cover. We have used a cloud fraction threshold of 20%, and data from the region below this threshold are only used. We have only used 15 random days from November for the

estimation. Because of this criterion, there is a reduction of 90.4% observations over the domain. Due to the monsoon overcast, the month of July is excluded since the number of cloud-free data is significantly low to deduce any meaningful results. As explained in Sect. 2.4.1, the modelled simulations are obtained by adding the corresponding representation error to the pseudo observations. Both the observation and simulation vector have 181320 (=1×12088×15) elements, and the state vector has only one element, which represents the scaling factor

of NEE fluxes over the entire domain. Posterior fluxes are calculated using the scaling factor, derived from minimizing the cost function (Eq. (2)).

### 3 Results and Discussions

#### 3.1 Agreement among global models

We first analyse the performance of current-generation global transport models in simulating $CO_2$ concentration

over Indian subcontinent. Note that a mere agreement among the coarse models is not sufficient to justify the models' performance over the region due to their large model errors in common and interdependency in terms of data sources. As mentioned in Sect. 2.2, we restrict this analysis to daytime-only values since different processes control the variability of $CO_2$ concentration at daytime and nighttime, and simulating nighttime variability is more complicated than the daytime (Lauvaux et al., 2009). For a consistent comparison among global models,

all the products are sampled at the same time for the region spanning from 67° E to 98° E and 7° N to 38° N. Figure 2a depicts the annual vertical profiles of $CO_2$ concentration, showing models' discrepancy in simulating the vertical gradients in concentration values including the boundary layer and the free troposphere. A notable difference is observed in the simulation of the gradient within the boundary layer by different models. The magnitude and the height up to which this positive gradient is observed is different for these models. LSCE

(both versions) has the largest positive gradient among these models (~1ppm). It shows the maximum concentration at around 700 m height and then a decrease in concentration. CarbonTracker also shows this positive gradient in the surface layers up to a height of 900 m. But the gradient is much smaller compared to the other two models. Among these four models, CarboScope does not exhibit this tendency in the lower atmosphere. Its concentration decreases linearly from the surface as the height increases.

The seasonal variability of $CO_2$ uptake through photosynthesis, release through ecosystem respiration, and the vertical transport is seen while analysing the monthly averaged $CO_2$ concentration profiles over Indian subcontinent (Figs. 2b and 3). Comparatively lower surface $CO_2$ concentrations are found during months with an active biosphere (June to October) than the rest of the period, owing to the more ecosystem productivity over Indian subcontinent in response to the availability of monsoon rainfall. Also, the presence of strong southwest

monsoon winds during June to September may result in bringing $CO_2$ depleted air from the southern hemisphere and thereby lowering the $CO_2$ concentration over the domain. Further, we see a $CO_2$ vertical profile with very small vertical gradient (~0.5 ppm within an altitude range of ~500 m to 4000 m) from June to October (Fig. 3). This is likely linked to the increased convective activities associated with the monsoon. The considerable inter-model variation in monthly averaged $CO_2$ concentration profiles as predicted by different global models is



problematic as it indicates the significant uncertainties in flux estimations over India. A part of this discrepancy
can come from the coarse resolution global model's inability to represent transport processes like convection
and vertical mixing, strength and distribution of anthropogenic sources and ecosystem activities that operate at
fine scales. The spatial distribution of $CO_2$ concentration shows structural differences among these models (see
Supplementary Fig. S1), indicating a substantial knowledge gap on accounting atmospheric $CO_2$ variability over
Indian subcontinent, which will have severe implications for the country's carbon budget estimations.

### 3.2 Representation errors in global transport models

As described in Sect. 2.3, we have calculated the representation error in the current generation global models
with a spatial resolution of $1° × 1°$, which allows us to identify the regions and processes with high variability in
$CO_2$ concentration. The larger the representation error, the larger are the variations that are caused due to sub-
grid processes within the grid box of $1° × 1°$. The spatio-temporal variability of representation error and the
influence of various factors in creating this variability are examined here.

### 3.2.1 Spatio-temporal patterns

Representation errors in the surface $CO_2$ concentrations for July and November are shown in Fig. 4. There is a
relatively high representation error in the coastal regions due to the temporal covariance between the coastal
meteorology and exchange fluxes. The $CO_2$ fluxes from coastal regions can be transported over the ocean and
accumulated in the shallow boundary layer over the ocean. The shallow boundary layer is a characteristic of the
marine atmosphere due to the less vertical mixing compared to land regions. Horizontal $CO_2$ gradients can also
be generated by the influence of highly varying biospheric fluxes under different advection patterns over the
land and ocean boundary. A similar mechanism is applicable to mountain regions where temporal covariance of
mountain-valley circulation and respired $CO_2$ fluxes are regulated by atmospheric radiation. We find that the
representation error for nighttime is characterized by high values compared to the daytime throughout the
analysed domain. This is expected due to the coupling between nocturnal shallow transport and different flux
processes, causing more local effects. During the nighttime, photosynthesis is absent, and respiration is the
major biospheric activity, leading to an increase in $CO_2$ concentration in the atmosphere. Together with the large
heterogeneity in flux distribution, mostly from respired $CO_2$ flux, the shallow boundary layer processes and
weak turbulence during nighttime cause $CO_2$ to be accumulated locally near the surface, showing large
variations.

In July, many low-pressure systems were active in the monsoon trough region (IMD weather reports,
https://mausam.imd.gov.in) whose presence influence the atmospheric transport (Li et al., 2021) and creates
representation error due to the associated mesoscale activity. The influence of the synoptic systems on $CO_2$
concentration can be observed during July (Fig. 5a) and November (Fig. 6b) with regions exhibiting well mixed
vertical gradients.. Strong mixing and vertical transport  associated with the low-pressure systems are visible
from these $CO_2$ concentration figures. Compared to July, we find higher representation error in November
owing to the wintertime transport with decreased vertical mixing and less biospheric uptake. A significant part
of this variation in November comes from the formation of low-pressure systems over the Bay of Bengal and
over the Lakshadweep area ($≈ 8° N, 74° E$) from 22 November onwards. One of these low-pressure systems in
the Bay of Bengal further developed and intensified as deep depression and moved to the southeast Arabian Sea





and evolved into a severe cyclonic storm (Ockhi) by 30 November. In addition to this, there persisted a western disturbance as an upper air cyclonic circulation over northern parts of Jammu and Kashmir during the period of

November 23-30.  Figure 6 shows the latitudinal averaged vertical cross-section atmospheric $CO_2$ concentration from the WRF-GHG simulation during two separate periods (period during days 11-20 and 21-30) in November. By comparing these two periods in November, we can see the influence of low-pressure systems on the $CO_2$ concentrations. Due to the impact of large horizontal and vertical flows associated with these synoptic systems, we have observed low $CO_2$ concentration values and high spatial variability in the eastern part of the

domain. These variations create significantly large representation errors compared to other periods during November, as will be discussed in more detail in section 3.2.2.

In the case of $XCO_2$, the magnitude of sub-grid scale variability is much smaller than that of surface $CO_2$ (Fig.7), but it follows a similar spatial pattern of surface sub-grid variations. This confirms the dominance of surface-level processes in causing sub-grid variability of column averages. The sub-grid scale variability in

$XCO_2$ reaches up to 3 ppm in some parts of the region especially where there are high variations in topographic features. For a large part of the domain, the representation error is found to be above 1 ppm, which is very close to or even larger than the typical precision of current satellite measurements ($\approx 1 - 1.5$ ppm) and it is capable of causing significant biases in the satellite inferred $CO_2$ fluxes over India. This result also reasserts the need for high resolution modelling over India when utilizing satellite observations.

Figure 8 shows the statistical distribution of the representation error during July and November, separated by daytime and nighttime. July shows a median representation error of 1 ppm and 1.5 ppm during daytime and nighttime respectively, while November shows a median value of 3 ppm and 3.5 ppm for daytime and nighttime respectively. During July, 95 % of the representation error is less than 3.2 ppm for daytime (6 ppm for nighttime) while it is 6 ppm for daytime (8 ppm for nighttime) in the case of November. For column average,

median values for representation error are 0.7 ppm and 1.3 ppm for July and November respectively. To further reduce the effect of random error that might be introduced by short-term weather phenomena, the correlated representation errors are calculated from the monthly averaged $CO_2$ field and is denoted as a systematic error ("Sys" in Fig. 8). Note that uncorrelated errors are expected to decrease when averaging over a sufficiently long period. As expected, the median representation errors for systematic representation errors are reduced for all

cases, showing the effect of random errors. Especially for November when the cyclonic event was present, the values for systematic errors (in the 95% percentile) for the surface $CO_2$ is considerably lower than total errors, reducing from 6ppm (daytime) and 8 ppm (nighttime) to 3 ppm (daytime) and 4 ppm (nighttime). In the case of column $CO_2$, this reduction is from 3.5 ppm (daytime) and 3.6 ppm (nighttime) to 1.5 ppm (daytime) and 1.6 ppm (nighttime) in the 95% percentile. Consistent with Fig. 8a both median and 95% percentile of the nighttime

representation errors are found to be higher than daytime except for November night.

We further carried out the estimation of representation errors for a spatial resolution of $0.5° \times 0.5°$ (Supplementary Figs. S2 and S3) to explore how much these variations can be resolved by a typical model operating at this spatial scale (e.g., regional models). On average, we find a decrease in representation errors to 0.78 ppm (~22 % reduction), and 1.6 ppm (~47% reduction) in the month of July (daytime) and November

(daytime) respectively. Exhibiting a similar spatial pattern of error for both resolutions of 0.5° and 1° indicates the need for a much finer modelling system than that of regional models with a typical resolution of 0.5° for



adequately representing the $CO_2$ dynamics. Though a reduction of representation error is observed for 0.5° compared to 1°, the emission hotspots and point sources are still pronounced, with high sub-grid scale variability, especially during nighttime. The above analyses indicate that the representation error alone can have significantly high values compared to the sampling errors (e.g., 0.1 ppm as per WMO standards for surface measurements), which demands higher resolution inversion systems for the optimal estimation of $CO_2$ fluxes.

### 3.2.2 Intra-month variability

Here we have used hourly representation errors to see its sub-monthly variations in July and November. Results show that daily variations in surface representation errors are minimal ($\pm$ 0.14 ppm) within a month for July, although there exists a clear distinction between daytime and nighttime values (existence of diurnal variability, see Fig. 9). Compared to other days in November, higher surface representation errors (~5 ppm) are found from 23 to 30 November in response to the synoptic systems prevailed. These results indicate that small variations in daily representation errors can be expected within a month. But noticeable variations with higher representation errors do happen in the occurrence of synoptic events like a cyclone. Such events need to be adequately represented in the model to capture the temporal changes in the regional spatial variability, which is difficult to be handled by global models because of the spatial-scale mismatches between model and synoptic events.

Interestingly, column representation error in July shows different behaviour compared to surface concentrations. We find comparatively higher (factor of more than two) representation error during 10 to 18 July than the rest of the period. This is due to the upper-troposphere dynamics associated with the development of low-level jet (LLJ) in the lower troposphere and the Tropical Easterly Jet (TEJ) in the upper troposphere during the monsoon season, as seen in Kottayil et al., (2019). Though the effect of LLJ and TEJ is visible throughout July (Fig. 5b), strong convective activity associated with the low-pressure systems is visible during July 10-18 (Fig. 5a). The combined effect of LLJ and TEJ can also be seen in the vertical distribution of the representation error (next section).

### 3.2.3 Vertical distribution

Figure 10 shows the vertical profile of representation error distribution within different bins of altitude. In addition to using 24 hours (denoted as "full-day"), we have also separated daytime and nighttime for the profile distribution. For November, we find the maximum representation error in the surface layer and most of the higher values are found to be within the lowest 4-6 km bins. Also, sub-grid scale variability decreases sharply with increasing altitude for November. This dominance of variability in surface concentration for November can be explained by surface flux heterogeneity influencing mole fractions in lower atmospheric layers (PBL) as described in van der Molen and Dolman, (2007) and Pillai et al., (2010). These dominances are found to be even more significant during the synoptic-scale event when the associated variations were mostly confined within the lower atmospheric levels (Supplementary Fig. S4). On the other hand, the representation error for July increases with altitude up to 12 to 14 km, showing the significant impact of meteorology above the boundary layer owing to the combined effect of LLJ and TEJ. Local convection can also produce variations in the upper level, but its scale of variations is expected to be less compared to the monsoon circulation systems during July. This implies that the monsoon circulations, along with extensive convective activities, can dominate the surface flux





heterogeneity in producing sub-grid variability, thereby causing significant biases in the flux estimations when

transport is not adequately accounted.

### 3.3. Influence of terrain heterogeneity on representation errors

The influence of terrain heterogeneity on representation error can be seen from the spatial figures in Figs. 4 and 7, where the large sub-grid scale variations are found in the Himalayan regions. Spatial variations in topography produce mesoscale circulation patterns and hence the variations in atmospheric $CO_2$ at fine scales. To further

explore the importance of using the high-resolution topography data on representing the $CO_2$ variability, we analyse the dependence of terrain variations (as derived from the standard deviation of terrain height) on representation error distribution. We have calculated the linear fit between topographic variability within the global climate models' grids and corresponding representation error to estimate the relation between them. Topographic variability within $1° \times 1°$ box is estimated as the standard deviation of topography (m) for all

$0.1° \times 0.1°$ boxes within the larger grid. Bins are created based on the values of this topographic variability, in which different points from different parts of the domain are binned together on the basis of their standard deviation of topography. Each bin is created with a size of 50 m variation in terrain height. The linear fit is estimated between the average value of topographic variability within a bin and the average value of representation error of the corresponding points in the bin. Our results show that the terrain heterogeneity alone

can explain about 53-62% of the surface representation errors over the domain (see Supplementary Fig. S5). In a similar way, we have estimated the influence of topographic variability in the formation of representation error in the column averaged model simulations. It is found that both surface and column representation errors show a very high correlation with the topographic variations in which topography alone can explain 81-89 % of representation errors in the column-averaged simulations. Though there exists a slight correlation between

surface flux heterogeneity (as derived from the standard deviation of VPRM-derived fluxes) and surface representation errors during daytime (11-35%), we find that this relationship is absent during nighttime, which is consistent with the findings from Pillai et al., (2010). This result underlines the need for using accurate Digital Elevation Models (DEMs) in the atmospheric transport models as one of the most critical datasets for determining the mesoscale atmospheric flows adequately.

### 3.4 Estimation of NEE flux uncertainty due to representation error

By following the assumptions and approach as given in Sect. 2.4, we have estimated the NEE flux uncertainty resulting from the representation errors. The results based on the OSSEs for nine observation sites are given in Table. 3. The scaling factors, which are calculated separately for each site by adjusting the prior fluxes using pseudo-observations, are applied to the VPRM fluxes. The total NEE flux for India estimated by VPRM for July

and November are -373.3 $MtCO_2$ per month and -417.1 $MtCO_2$ per month respectively. The flux uncertainties over India that arise solely due to the contribution from the representation error are estimated to be 54.4 to 60.8 $MtCO_2$ per month (14.5% to 16.2%) for July and 26.2 to 31.4 $MtCO_2$ per month (6.3% to 7.5%) for November while utilizing data from nine observation stations. July showed the maximum flux uncertainty due to the enhanced biosphere activity and unresolved convection activities. As expected, the flux uncertainty is higher

when using nighttime values for both months. In the same manner, as explained in Sect. 2.4.2, we have estimated the flux uncertainty while using satellite observations. We have only estimated the NEE flux





uncertainty during daytime in November in accordance with the availability of satellite observations over the domain. The scaling factor is calculated for the entire domain where the cloud fraction is less than 20%. The total flux uncertainty, while using satellite observations for November over India, is 39.2 $MtCO_2$ per month
(9.4%). The estimated uncertainties are considerable for the carbon budget assessment especially given that these errors are arising solely from the global models' representation error. Note that calculated representation error does not include other transport error sources such as advection, convection or vertical mixing.

**3.5 Possible treatment of representation error in the global model.**

The simplest possible way to minimize the uncertainty in flux estimation using a coarse model is to construct a
parameterization model that can take to the account of representation error using explanatory variables. For this, we assert a multivariable model to capture spatial patterns in the representation error. The hypothetical satellite observations are used for this analysis. The linear model using only one explanatory variable, terrain heterogeneity, as derived from the previous analysis in Sect. 3.3, captures the derived representation error in the southern part of India with less than 1 ppm bias (see Supplementary Fig. S6). However, in the northern part of
the region, especially in the Indo Gangetic Plain (IGP) and Rajasthan, we find that the terrain heterogeneity alone cannot explain the spatial patterns of the representation error, and the bias is larger than 1 ppm. Our further analysis using a multivariable linear model with explanatory variables that include sub-grid variations of terrain, biospheric and anthropogenic fluxes shows a remarkable improvement in representing the derived representation error all over the domain with a $R^2$ value of 0.68 (Fig. 11). The bias between the modelled and
derived representation error is found to be well below 1 ppm in most parts of the domain. By employing this model, we see that the total inferred flux uncertainty, while using satellite observations for November over the entire domain, is reduced from 9.4% (39.2 $MtCO_2$ per month) to 2.2% (9.1 $MtCO_2$ per month). This finding provides a possibility for a parameterization that can be further developed in inverse models or data assimilation systems, which defines the degrees of freedom for describing the posterior states. However, note that this simple
parameterization may not be useful when there exists large influence from monsoon circulations as seen for July (see Sects. 3.2.1 and 3.2.3). Applying this parameterization scheme to the specific problem requires a high-resolution map of the terrain and prior information on anthropogenic and biogenic fluxes. The uncertainties in the topography can significantly impact flux estimation, and the likely reduction of flux uncertainty depends on the accuracy of the DEM employed. The caveat of this linear model is that the uncorrelated spatial variability in
the prior and true states of the fluxes is ignored in the present form, which cannot be the case for the real inverse problems. This assumption obviously hampers the system in achieving the maximum reduction in uncertainty, and further study is needed to refine this model from a practical perspective. We emphasise, however, that the above parameterization does not require high-resolution simulation of transport, which has high computational costs.

**4. Conclusion**

Given the upcoming availability of atmospheric observations over India, significant effort is required to critically enhance the modelling capabilities to derive carbon budgets over India within the definite confidence intervals and at scales relevant to ecosystem and countrywide policy-making. The misrepresentation of transport phenomena and unresolved flux variations in the modelling system operating at coarse grids hinders the



effective utilization of observations. In this context, quantification of the spatial variability of atmospheric $CO_2$ mixing ratio and assessment of its seasonal dependence are necessary to make optimal use of observations in the inverse framework. In this study, the atmospheric $CO_2$ spatial variability is represented using the WRF-GHG model at 10 km spatial resolution. While investigating the shortcomings of the current-generation model on quantifying regional sources and sinks of $CO_2$, the present study demonstrates the potential of implementing

a high-resolution atmospheric $CO_2$ transport modelling framework over India. The importance of horizontal resolution in the models and the adequate representation of topography and flux variations in capturing $CO_2$ variability is given a focus here, by quantifying the representation error resulting from the coarse model's inability to resolve the fine-scale variations accurately.

Through our inter-model comparisons, we quantify the variability in the $CO_2$ mole fractions predicted by the

state-of-the-art global models (CarboScope, LSCE and CarbonTracker) in terms of their spatial and vertical variations even at a monthly scale. The large model spread among the vertical profiles (~1 ppm) and differences in vertical gradient indicate a severe knowledge gap in the estimations of fluxes even at the global scale. It can be argued that a significant part of these differences arises due to the lack of observational constraints over India, which leads to a possible compensatory model artefact over this region in order to match the global mass

constraint. Noteworthy is that none of these models includes ground-based data from the Indian subcontinent. At the same time, it is also expected that the spatial variability of the observed atmospheric $CO_2$ mole fractions can be large so that these coarse models fail to represent them adequately, thereby leading the inverse optimization to infer the state which is not close to the truth as is required in regional $CO_2$ budget for various applications.

By quantifying the unresolved variation of the global models with a spatial resolution of $1° \times 1°$, overall, we

find that the representation error can be up to ~10 ppm for the surface $CO_2$ and about 3 ppm for column-averaged values, which are markedly larger than the sampling errors. For example, the achievable precision of satellite-based $XCO_2$ is around 1 ppm for current satellites: OCO2 and GOSAT (Liang et al., 2017; O'Dell et al., 2012; Wunch et al., 2017). In the case of high accuracy in situ measurements, the typical uncertainty for $CO_2$ measurements is less than 0.1 ppm (Andrews et al., 2014). These sampling errors are significantly smaller

than the size of the corresponding model's representation error. This indicates that the employed models need to be critically improved in terms of capturing mesoscale phenomena and fine-scale flux variability in order to maximize the potential of deducing the information obtained from these high precision measurements, thereby improving the estimation of surface fluxes. The mesoscale features and associated $CO_2$ gradients can be generally captured by increasing the spatial and temporal resolution of the transport models. However, by

merely increasing the resolution without having a realistic representation of terrain heterogeneity and flux (both natural and anthropogenic) variability would not be beneficial. Note that the uncertainties in the high-resolution fluxes can worsen the model's skills, whose effect would not be more pronounced at coarser resolutions due to the diffusive nature of fluxes, as seen in Agustí-Panareda et al. (2019).

Although the magnitude of the sub-grid variability of the total column is an order smaller than the variability at

the surface, the spatial pattern remains similar for both, owing to the dominance of surface heterogeneity in topography and fluxes. Our results are consistent with Pillai e al. (2010), who show that there exist regional differences in the sub-grid variability for both surface and column $CO_2$. Coastal sites and mountain are pronounced with high representation error ($\approx$6 ppm), while emission hotspots are distinguishable with



variability as large as ≈8 ppm. Larger values are typically associated with the nocturnal shallow boundary layer

dynamics and the stronger respiration signal with considerable flux variability.

We identify the seasonal changes in the sub-grid variability over India through our analysis during monsoon and post-monsoon periods. November shows a median representation error of 3 ppm during daytime which is three times more than the median representation error for July. The derived seasonal differences in structural patterns of the sub-grid variability facilitate to (1) quantify what would be transport errors associated with incorporating

seasonally varying observations (both surface and column observations) into atmospheric transport models when the model resolution differs from the spatial scales of observations (which reflects in the confidence intervals of the inferred $CO_2$ fluxes via inverse modelling) (2) determine what drives the seasonality in sub-grid variability and ultimately (3) design the possible parameterization of representation error with a seasonal component in the inverse modelling framework as well as identify periods (or seasons) where the use of this

parameterization would be valid to improve our estimations of $CO_2$ fluxes. Further, the seasonal spatial variability analysis of column averages can provide useful information for the satellite community to gap-fill the satellite soundings over India when large data gaps and low sounding precision on daily or monthly time scales are present, which is especially the case for monsoon periods in India. For example, the impact of the combined effect of LLJ and TEJ associated with monsoon can be seen in the vertical distribution of sub-grid variability for

July. During this period, the average representation error increases to ~3 ppm with altitude up to 12 to 14 km, in contrast to the post-monsoon period in which representation error decreases sharply with increasing altitude. This sharp decrease of sub-grid variability with altitude is expected and is consistent with (Pillai et al., 2010; van der Molen and Dolman, 2007). The results for the monsoon period assert the need for high resolution modelling over India incorporating both upper and lower troposphere. The cyclonic storm (Ockhi) in November

generates considerably more $CO_2$ spatial gradients than the rest of the period, which increases the sub-grid variability of $CO_2$ by ~ 1 ppm on average. The impact of cyclonic activities on $CO_2$ distribution and their frequent occurrence over the Indian subcontinent reassert the need to employ the models at high resolution for capturing the fine-scale $CO_2$ variability that coarser-resolution models cannot resolve. Increasing the model's resolution to $0.5° × 0.5°$ has shown an improvement in capturing variability with representation error reduction

of 22% and 47% for summer time and winter time respectively. By showing the existence of considerable magnitude of unresolved variability in 0.5° sub-grid scale with a similar spatial pattern of error as of 1° spatial resolution, we demonstrate the need for a much finer resolution than 0.5° for representing the atmospheric $CO_2$ over India.

The inaccuracies, as mentioned above, are large enough to manifest themselves in transport model biases. We

construct a simple inversion example to illustrate the impact of derived representation error on estimating the fluxes. In the light of the upcoming network of observations, we use pseudo-observations from nine observatories over India and hypothetical satellite observations. With the underlying assumptions, the total flux uncertainty solely due to the unresolved sub-grid variations is estimated to be 6.3 to 16.2% of the total NEE while utilizing data from nine observation stations over India. Similarly, the total expected flux uncertainty,

while using satellite observations for November over the entire domain is estimated to be 9.4%.

Further, the potential of a simple parameterization scheme has been presented to understand the origin of the representation error and take into account its effects in a global model. For this case, we have shown that a



multivariable linear model with explanatory variables of sub-grid variations of terrain, biospheric and anthropogenic fluxes can help to simulate the expected representation error in the global model. Employing this

linear model in a global model would thus redefine the likelihood of better estimates (improving the state of knowledge) with variance greater than that of the measurement error in the inverse framework by minimizing the modelling error. In our specific example, on utilizing satellite observations for November, we see a significant reduction in the total flux uncertainty from 9.4% to 2.2% after applying the linear model. The proposed method is easy to implement in the coarse models as it does not require computationally expensive

transport simulations at high-resolution. However, more work is needed to demonstrate the extent of applicability of this method to minimize the flux uncertainties while utilizing actual observations. As we see a significant dependence of the distribution of sub-grid variability on terrain variations, our results reinforce the requirement for using accurate DEMs in the atmospheric transport models to accurately determine the mesoscale atmospheric flows. The terrain heterogeneity alone can explain about 53 to 62% of the surface

representation errors over the domain. At the same time, we also note that such a simple parameterization as proposed here might not be adequate to capture the key aspects of the sub-variability when there exists large influence from monsoon circulations.

Overall, our results show that the mesoscale transports associated with seasonally reversing monsoon systems and flux variability contribute to fine-scale variations that the current-generation models cannot resolve. By

employing a high-resolution model that relies on the resolution of transport and surface fluxes as well as the resolution of the topography, our results provide a baseline for overcoming the shortcomings as mentioned above and accounting for the realistic distribution of atmospheric $CO_2$.





**Code/Data availability**

The WRF-Chem source code is publicly available at https://ruc.noaa.gov/wrf/wrf-chem/ (last access: 10 August 2019). The CarbonTracker (CT-2019B) products are available online at http://carbontracker.noaa.gov (last access: 21 July 2020, Jacobson et al., 2020). The data from the CarboScope inversion system are available online at http://www.bgc-jena.mpg.de/CarboScope/ (last access: 20 July 2020, Rödenbeck et al., 2003). The data from the LSCE modelling system used in this study are available at http://atmosphere.copernicus.eu (last access:

22 July 2020, Chevallier et al., 2019). The sub-grid variability products based on the WRF-GHG model simulations can be accessed from https://zenodo.org/record/4781938 (last access: 23 May 2021, Thilakan and Pillai, 2021). The WRF-GHG model $CO_2$ simulations used for this study are available upon request to the corresponding author, Dhanyalekshmi Pillai (dhanya@iiserb.ac.in, kdhanya@bgc-jena.mpg.de). The EDGAR data used in this study are publicly available at https://edgar.jrc.ec.europa.eu/ (last access: 15 March 2020,

Crippa et al., 2018). The GFAS data are publicly available at http://apps.ecmwf.int/datasets/data/cams-gfas/ (last access: 15 March 2020, Kaiser et al., 2012). The ERA5 data are available at https://cds.climate.copernicus.eu/cdsapp#!/home (last access: 18 March 2020, Hersbach et al., (2020)).

**Author Contribution**

DP designed the study and performed the model simulations. VT performed the analysis and wrote the paper. VT and DP interpreted the results. CG, MG, AR and TAM provided significant input to the interpretation, and the improvement of the paper. All authors discussed the results and commented on the paper.

**Competing interests**

The authors declare they have no conflict of interest.

**Acknowledgements**

This study is supported by the funding from the Max Planck Society allocated to the Max Planck Partner Group at IISERB and the Science and Engineering Research Board (SERB) through Early Career Research Award (ECR/2018/001111) to DP. TAM acknowledges the financial support provided by SERB grant (Junior Research Fellowship). We acknowledge the support of IISERB's high performance cluster system for computations, data

analysis and visualisation. The WRF-Chem simulations were done on the high performance cluster Mistral of the Deutsches Klimarechenzentrum GmbH (DKRZ).



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

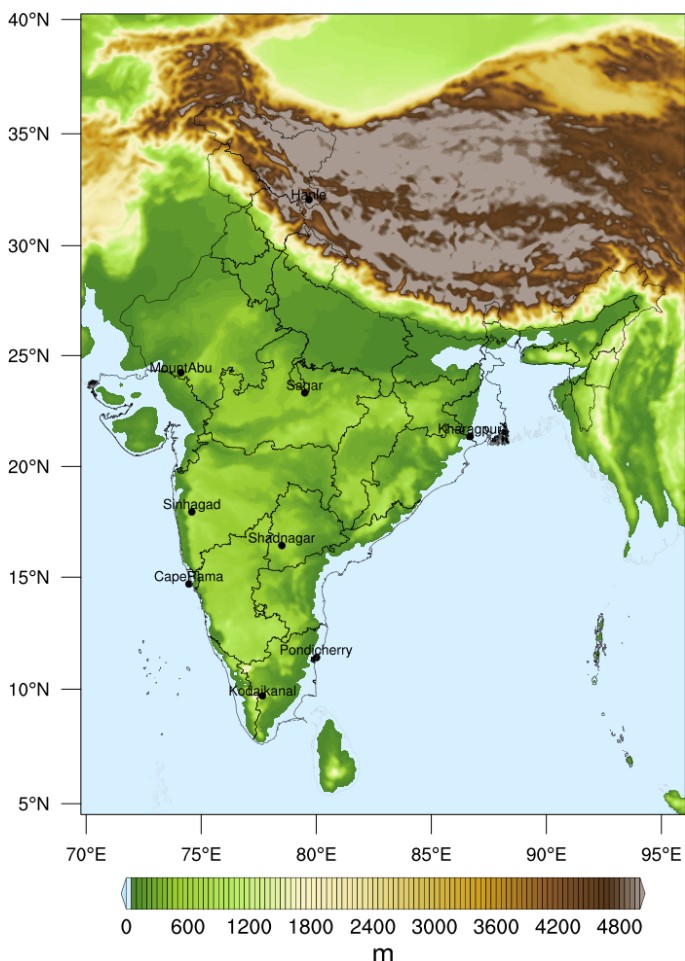


**Figure 1: The WRF-GHG model domain used in this study, showing topography. The CO$_2$ monitoring sites over India used for the OSSE experiments are marked. Not all these observation stations are currently fully operational. The colour scale is restricted to 5000 km for the better visualization of terrain details over the Indian subcontinent.**


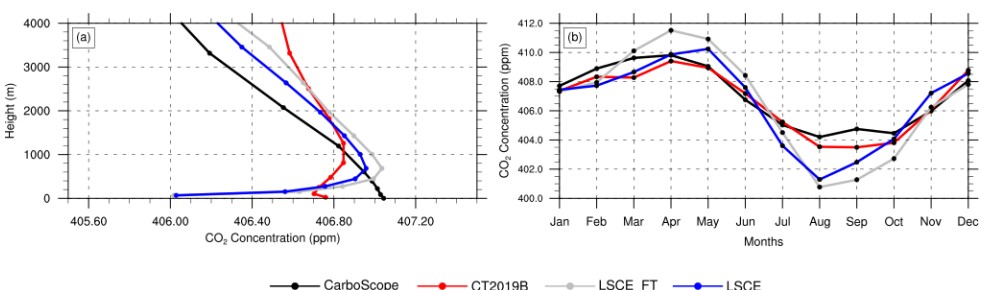

**Figure 2: Comparison of global models over the model domain during daytime (11:30 to 16:30 local time) in 2017. a) Annually averaged vertical profiles of $CO_2$ concentration in the lower troposphere b) Time series of monthly averaged $CO_2$ concentration at surface (~100 m above surface).**


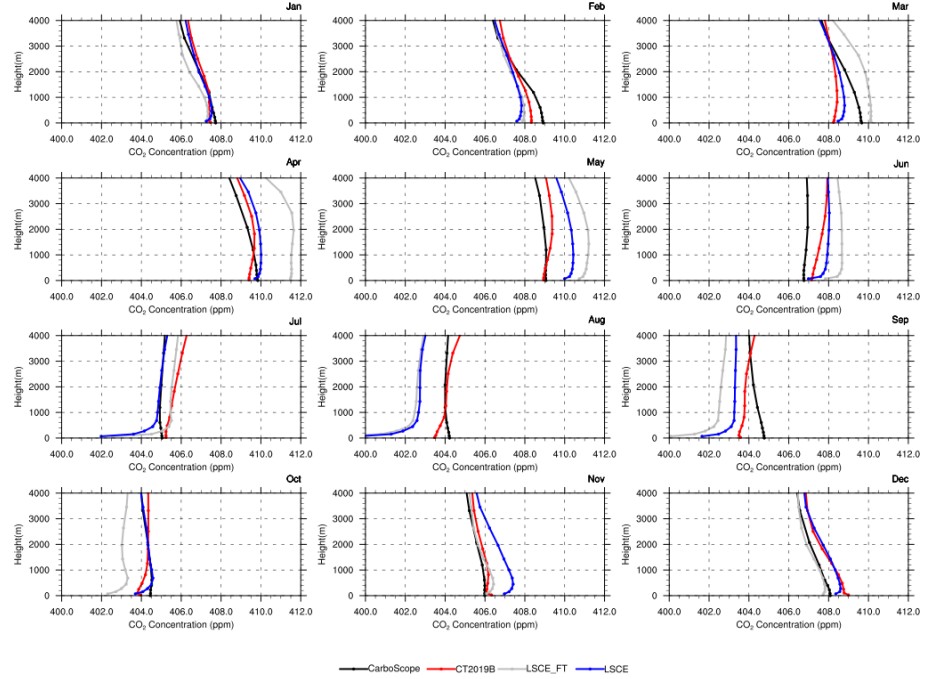

**Figure 3: Comparison of average monthly vertical profiles of $CO_2$ concentration from global atmospheric transport models over the model domain during daytime (11:30 to 16:30 local time) in 2017. Panels show data for respective months as indicated on the top of each panel.**




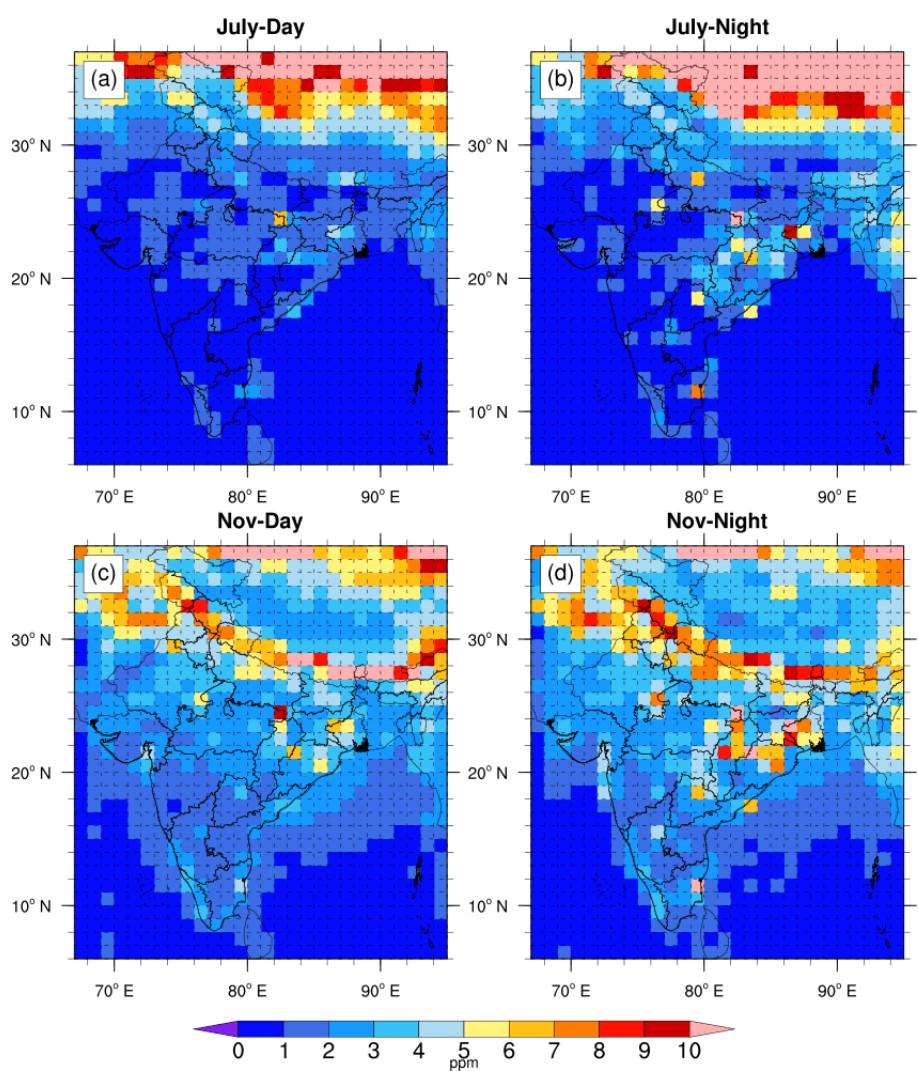

**Figure 4: Monthly averaged values of representation error estimated for surface CO$_2$ concentration (second model level ~200 m) during 2017. a) July daytime (11:30 to 16:30 local time) b) July nighttime (23:30 to 4:30 local time). c) November daytime. d) November nighttime.**




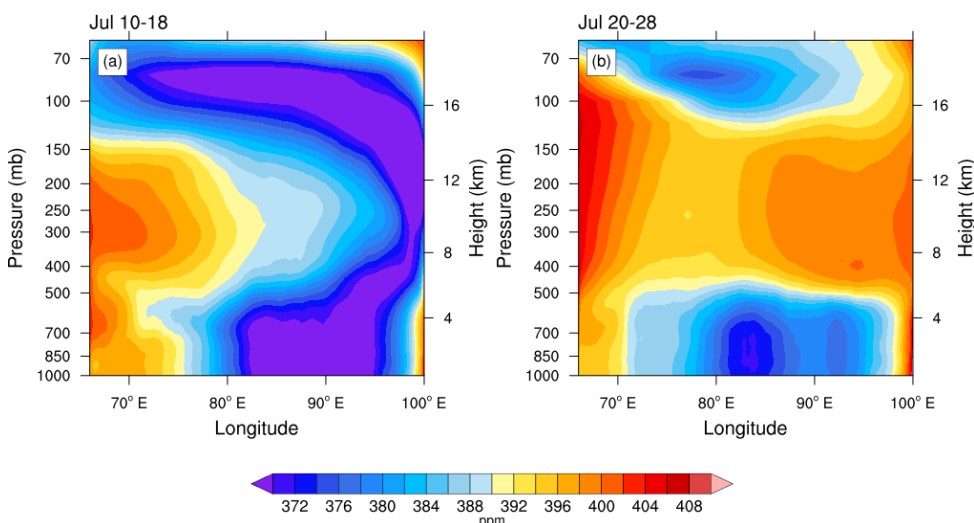


**Figure 5: Latitudinal averaged vertical cross-section of CO$_2$ concentration from WRF-GHG simulations during July 2017. a) Vertical cross-section of CO$_2$ concentration during July 10-18. There are a number of low-pressure systems present over the monsoon-trough region during this period b) Vertical cross-section of CO$_2$ concentration during July 20-28. Low-pressure systems are not so pronounced during this time. The effect of LLJ and TEJ is visible**

**throughout the whole month.**

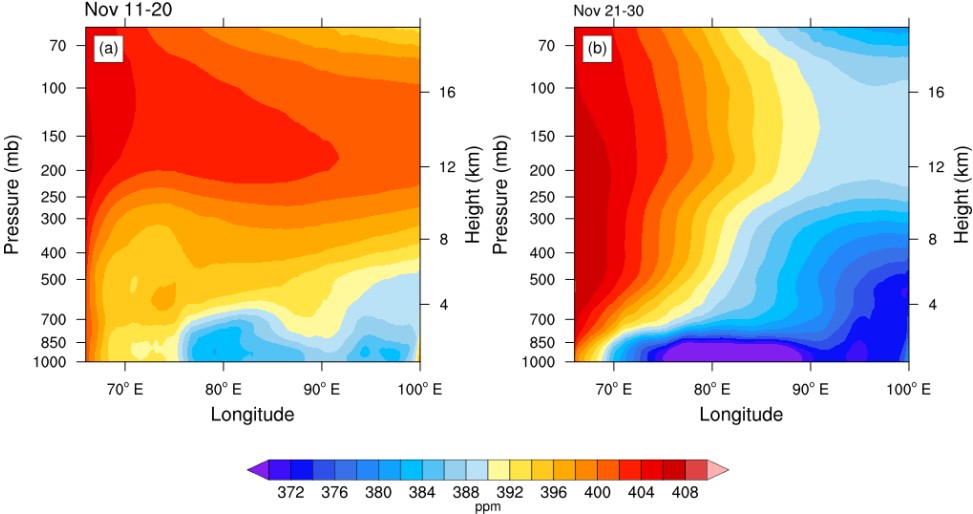

**Figure 6: Latitudinal averaged vertical cross-section of CO$_2$ concentration from WRF-GHG simulations during**

**November 2017. a) Vertical cross-section of CO$_2$ concentration during November 11-20. There are no low-pressure systems present during this period. b) Vertical cross-section of CO$_2$ concentration during November 21-30. This period is characterized with low-pressure systems in Bay of Bengal and Arabian sea, one of which intensified further and formed into the severe cyclone Ockhi.**


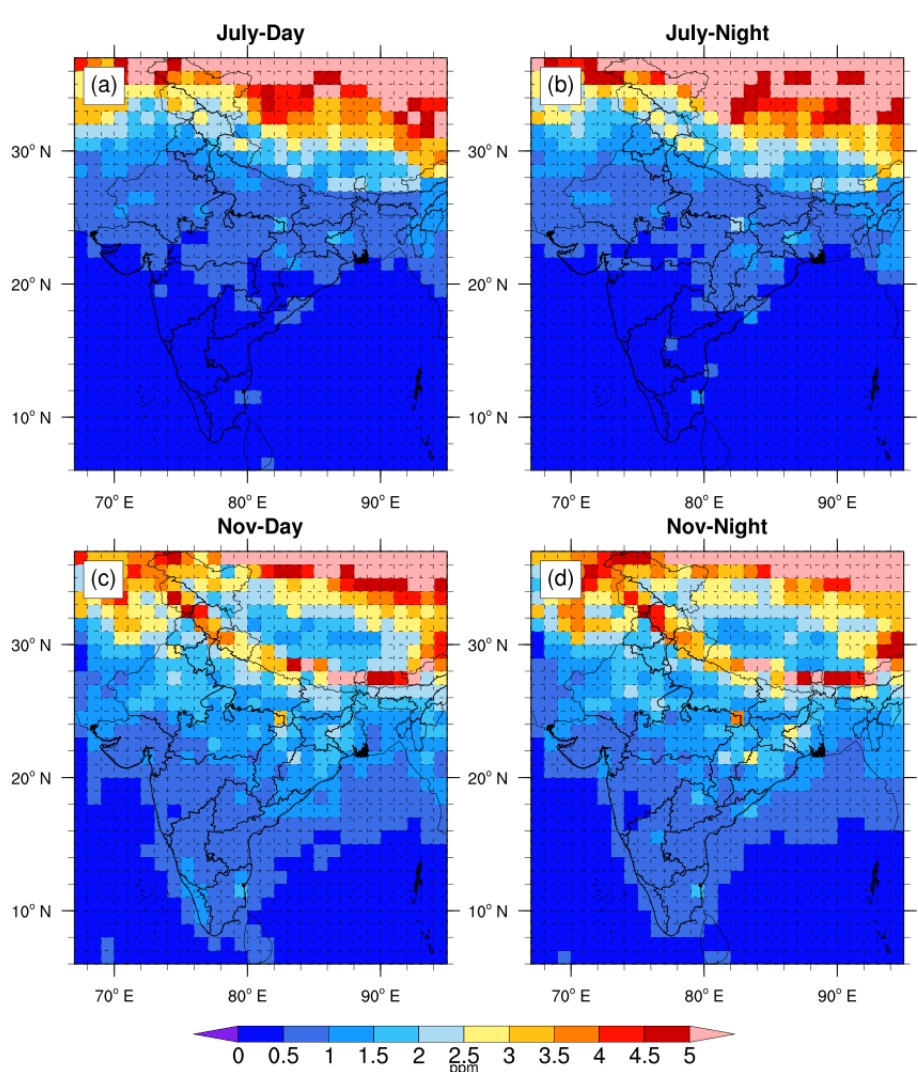

**Figure 7: Monthly averaged values of representation error estimated for column averaged $CO_2$ concentration during 2017. a) July daytime (11:30 to 16:30 local time) b) July nighttime (23:30 to 4:30 local time). c) November daytime. d) November nighttime.**







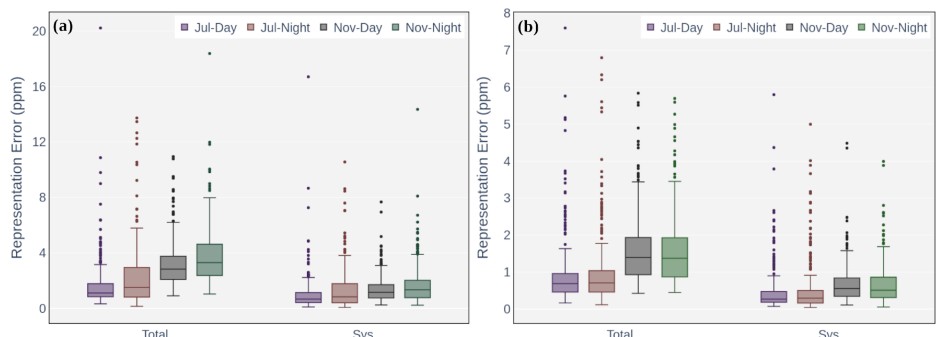

**Figure 8: Variability of derived representation error over India. Boxes indicate the central 50%, the bar across the box is the median value, and the whiskers indicate the values between 5 and 95 percentiles. Individual data points shown are the outliers. a) Representation error estimated for the surface $CO_2$. b) Representation error estimated for the column averaged $CO_2$. The notations, "Total" indicates the total error containing systematic and random errors and "Sys" represents systematic errors only.**




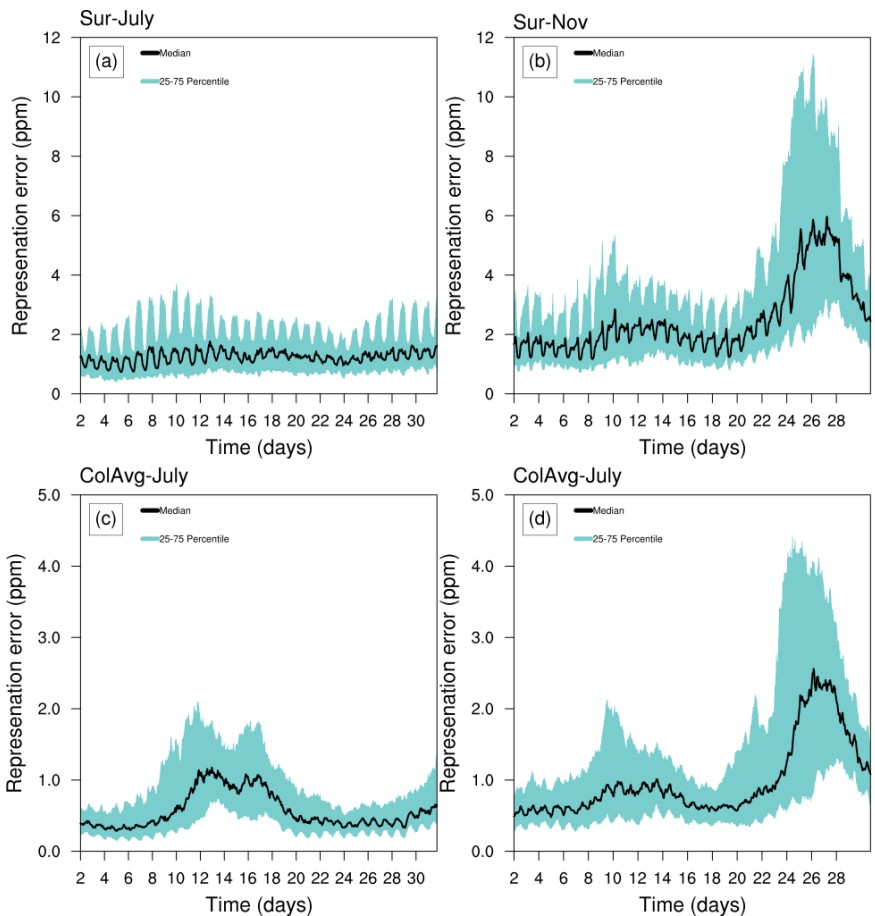

**Figure 9: Intra month variability of representation error over India. a) Variability of surface representation error during July 2017 b) Variability of surface representation error during November 2017 c) Variability of column averaged representation error during July 2017 d) Variability of column averaged representation error during November 2017. Median values are plotted with black curves and a shaded region indicates 25 to 75 percentiles of data.**





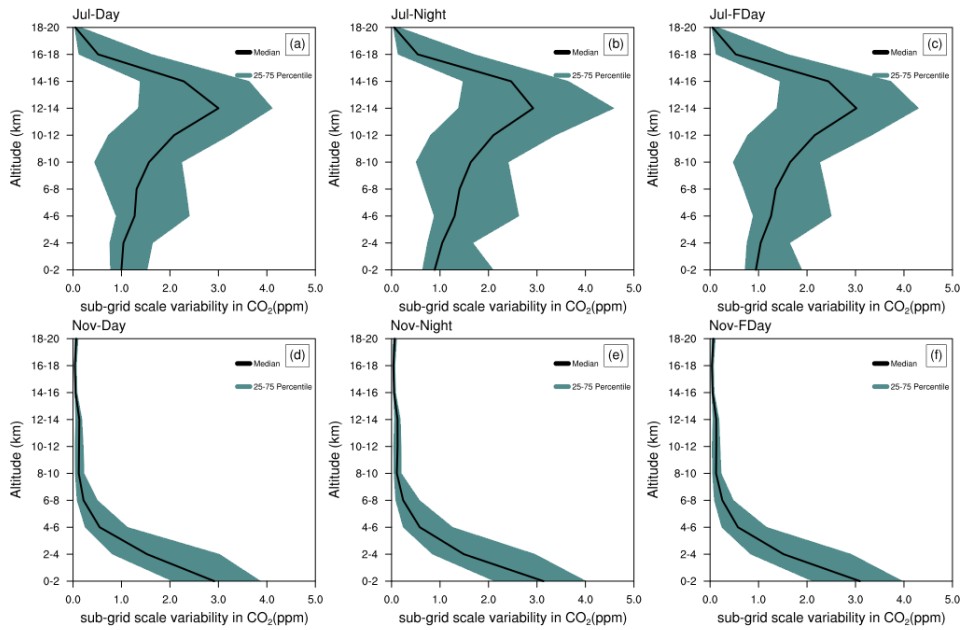

**Figure 10:  Variability of representation error over India with altitude for July and November 2017. a) July daytime, b) July nighttime, c) July full time, d) November daytime, e) November nighttime, and f) November full time. Median values are plotted with black curves and the shaded region indicates 25 to 75 percentiles of data.**





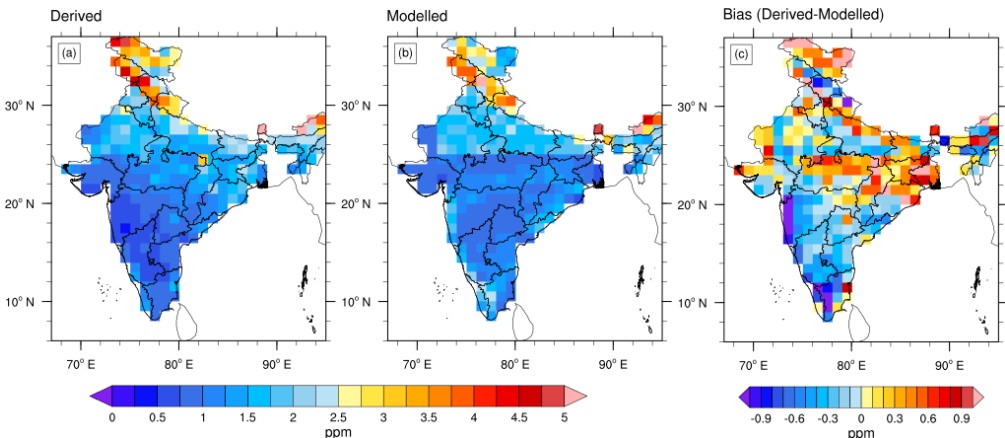

**Figure 11: Monthly averaged values of representation error estimated for column averaged $CO_2$ concentration during daytime (11:30 to 16:30 local time) in 2017. a) Representation error derived from WRF-GHG simulations as explained in Sect. 2.3. b) Representation error calculated from the multivariable linear model as described in Sect. 3.5. c) Bias of multivariable linear model (difference between (a) and (b)).**



**Table 1: WRF-GHG Model setup**


| Domain | | | | | | |
|---|---|---|---|---|---|---|
| Configuration | Single domain with horizontal resolution of 10 km; 39 vertical levels; 307 × 407 grid points | | | | | |
| Vertical coordinates | Terrain-following hydrostatic pressure vertical coordinates | | | | | |
| Basic equations | non-hydrostatic; compressible | | | | | |
| Grid type | Arakawa-C grid | | | | | |
| Time integration | 3rd order Runge-Kutta split-explicit | | | | | |
| Spatial integration | 3rd and 5th order differencing for vertical and horizontal advection respectively; both for momentum and scalars | | | | | |
| Timestep | 60 s | | | | | |
| **Physics schemes** | | | | | | |
| Radiation | Rapid Radiative Transfer Model (RRTM) for Longwave & Dudhia for shortwave | | | | | |
| Microphysics | WSM 3-classic simple ice scheme | | | | | |
| PBL | YSU | | | | | |
| Surface layer | Monin-Obukhov | | | | | |
| Land-surface | NOAH LSM | | | | | |
| Cumulus | Grell-Devenyi ensemble scheme | | | | | |
| **Emission fields** | | | | | | |
| Flux type | Product | Version | Spatial resolution | Temporal resolution | Source/website | Reference |
| Anthropogenic | EDGAR | v4.3 | 10km | Annual | https://edgar.jrc.ec.europa.eu/ | Crippa et al., (2018) |
| Biomass burning | GFAS | v1.2 | 10km | Daily | http://apps.ecmwf.int/datasets/data/cams-gfas/ | Kaiser et al., (2012) |
| Biospheric | VPRM | | Adapted to model | Adapted to model | | Mahadevan et al., (2008) |
| **Initial and Lateral Boundary conditions** | | | | | | |
| Field | Product | Version | Spatial resolution | Temporal resolution | Source/website | Reference |
| Meteorology | ERA5 | n/a | 25km | 1hour | https://cds.climate.copernicus.eu/cdsapp#!/home | Hersbach et al., (2020) |
| Tracer | ECMWF/CAMS | gqiq | 50km | 6hour | http://atmosphere.copernicus.eu | Agustí-Panareda et al., (2019) |





**Table 2: Specifications of the different global model products used in this study**

| Data availability | | | | | | |
|---|---|---|---|---|---|---|
| Product | Version | Spatial resolut-ion | Vertical levels | Temporal resolution | Source/website | Reference |
| Cabron Tracker | CT2019B | 3 x 2 | 25 | 3 hours | http://carbontracker.noaa.gov | Jacobson et al., (2020) |
| CarboScope | v2020 | 5x3.8 | 19 | 6hours | http://www.bgc-jena.mpg.de/CarboScope/ | Rödenbeck et al., (2003) |
| LSCE | v18r3 | 3.7x1.8 | 39 | 3hours | http://atmosphere.copernicus.eu | Chevallier et al., (2019) |
| LSCE | FT18r1 | 3.7x1.8 | 39 | 3hours | http://atmosphere.copernicus.eu | Chevallier et al., (2019) |

| Data used in the inverse model simulations | | | | | | | |
|---|---|---|---|---|---|---|---|
| Product | Version | Forwa-rd Model | Observa-tion data | Anthropoge-nic emission fields | Biospheric emission | Fire emission | Oceanic emission |
| Cabron Tracker | CT2019B | TM5 | Ground based | Miller and ODIAC | CASA | GFED and GFED CMS | OIF and Takahashi et al., (2009) |
| CarboScope | v2020 | TM3 | Ground based | EDGAR EDGAR, | LPJ Biosphere Model | CDIAC | SOCAT |
| LSCE/PyVar | v18r3 | LMDz 6A | Ground based | CDIAC and GCP | ORCHIDEE 4.6.9.5 | FED and GFAS | Denvil-Sommer et al., (2019) with updates described in Friedlingstein et al., (2019) |
| LSCE/PyVar | FT18r1 | LMDz 6A | Satellite (OCO-2 NASA) | EDGAR, CDIAC and GCP | ORCHIDEE 1.9.5.2 | GFED and GFAS | Landschutzer et al., (2018) |




**Table 3: Flux uncertainty over India calculated from the OSSE experiments using pseudo-observation network of surface observations. The time Filter indicates the time of the data sampled for estimation of the scaling factors. Full day – 24 hours in each day; Day – 11:30 to 16:30 local time; Night – 23:30 to 4:30 local time. * The fraction of uncertainty to the total NEE.**

| Month | Time Filter | True flux, $\Phi_{true}$ (MtCO$_2$ per month) | Flux uncertainty, $\bar{S}_{rep}$ (MtCO$_2$ per month) In brackets: fraction of uncertainty* (%) |
|---|---|---|---|
| July | Full Day | -373.31 | 46.35 (12.4) |
| July | Day | -373.31 | 54.40 (14.5) |
| July | Night | -373.31 | 60.84 (16.2) |
| November | Full day | -417.12 | 28.62 (6.8) |
| November | Day | -417.12 | 26.28 (6.3) |
| November | Night | -417.12 | 31.41 (7.5) |
