# Peer review of "Towards monitoring CO2 source-sink distribution over India via inverse modelling: Quantifying the fine-scale spatiotemporal variability of atmospheric CO2 mole fraction"

_Atmospheric Chemistry and Physics, 2021_

## Editor Comment (EC1)

**Comments on "Towards monitoring CO$_2$ source-sink distribution over India via inverse modelling: Quantifying the fine-scale spatiotemporal variability of atmospheric CO$_2$ mole fraction"**

**Vishnu Thilakan, etal., ACP (under review)**

This is a very interesting and timely paper for the regional carbon cycle and flux inversions. The authors have focused to obtain representative errors in the models leading to significant errors in the source/sink estimations in the inversions. Notably, they have quantified that these REs can be as large as 9 ppm which is well above the observational uncertainty and as comparable to enhancement at any point source region.

Unfortunately, the study fails to acknowledge recent other similar studies in the Indian flux inversions. To be specific, Nalini et al., (2019) quantified the potential uncertainty reduction achievable by using data from existing tower-based observations over India. Moreover, they also have proposed 17 new stations spread across various parts of India and put forward this important recommendation to initiate observations to benefit Indian flux inversions especially when satellite constellations fail to capture Indian footprints during the heavy rainy season. A large ensemble of particle trajectories and Bayesian inversions with incremental optimization is done in their OSSE work. It is pity that the authors fail to discuss this paper in their work.

Tiwari et al, (2013), Ravi Kumar et al., (2016) also discussed the atmospheric concentration variability over India at seasonal and intra-seasonal scales. The authors also discuss the similar RE variation at seasonal and inter-monthly scale due to changes in meteorology. Therefore the above papers should be mentioned in this context.

**References**
Nalini, K., S. Sijikumar, Vinu Valsala, Yogesh K. Tiwari, Radhika Ramachandran, 2019, Designing surface CO2 monitoring network to constrain the Indian land fluxes, Atmospheric Environment 218 2019. 117003, https://doi.org/10.1016/j.atmosenv.2019.117003, 28 Sept., 2019

Ravi Kumar K., Valsala V., Yogesh K. Tiwari*, Revadekar J.V., Pillai P., Chakraborty S., Murtugudde R., 2016, Intra-seasonal variability of atmospheric CO2 concentrations over India during summer monsoons, *Atmospheric Environment*, 142, October 2016, ISSN:1352-2310, DOI:10.1016/j.atmosenv.2016.07.023, 229-237

Yogesh K. Tiwari, Vinu Valsala, Ramesh K. Vellore, Ravi K. Kunchala, 2013, Effectiveness of surface monitoring stations in capturing regional CO2 emissions over India. *Climate Research*, Vol. 56: 121–129, 12 March 2013, ISSN 1616- 1572, DOI: 10.3354/cr01149

---

## Author Comment (AC1)

**Authors' response to Referees**

We greatly appreciate both referees for providing insightful comments on our manuscript. We have addressed all the comments, suggestions and concerns raised by the referees, and incorporated the associated modifications in the manuscript. We believe that the manuscript has improved significantly after including these changes. Thank you.

Our responses and modifications in response to the reviewers' comments are listed below: the reviewers' comments are given in regular black font, our responses are given in regular blue font, and the changes in the revised version are given in italic blue font.

Thank you

Authors

**Reviewer 1**

Accurate assessment of sources and sinks of CO2 is essential in planning and implementing the mitigation strategies for greenhouse gas emission and associated climate change. In this study using inverse modelling it is demonstrated that there is a need for implementing a high-resolution modelling framework over the Indian subcontinent to better understand processes regulating CO2 sources and sinks.

This is a very interesting and important study, which merits its publication in ACP. The scientific content, the quality of the study and its presentation is good, however in some parts the text is very descriptive and technical. I suggest some minor revisions before publication by ACP.

Thank you for appreciating the importance and merit of our work. We have done the revisions in the manuscript as per the suggestions.

In general my impression is that the conclusion section is very long and to many things are discussed within. I propose to shorten the conclusion to be more condensed and to focus to the main results.

We agree. The conclusion is revised accordingly. Please see the revised manuscript.

**Specific comments:**

P1/L21: `We show that the unresolved variability in the coarse model reaches up to a value of 10 ppm at the surface, which is considerably larger than the sampling errors, even comparable to the magnitude of mixing ratio enhancements in source regions.'

What is the meaning of 'mixing ratio enhancements'? The regional variability of monthly mean surface CO2 concentration in India as shown in Fig. 4.? Further, the variability of CO2 time series of monthly averaged CO2 concentration at surface as shown in Fig. 2b are also in the range of 10 ppm, that could be mentioned here.

By the term 'mixing ratio enhancement in source regions', we refer to the increment in  $CO_2$  concentration in a region from background values, in association with the presence of anthropogenic or natural sources. To make it clearer, we have edited the manuscript and also included the range of seasonal variability as shown in Fig. 2b.

The text is modified as follows:

L19-25: "We show that the representation error due to unresolved variability in the coarse model at a horizontal resolution of one degree (~ 100 km) reaches a median value of 3 ppm and 1.3 ppm for the surface and column  $CO_2$  respectively, which are considerably larger than the measurement errors. The extent of monthly averaged surface representation error reaches up to ~10 ppm depending on season, which is even comparable to the magnitude of seasonal variability or concentration enhancement due to hotspot emissions. Representation error shows strong dependence on multiple factors such as time of the day, season, terrain heterogeneity, and changes in meteorology and surface fluxes."

P2/L91: 'The monsoon convection that transports the boundary layer air into the free atmosphere (mainly to the upper troposphere and to the lower stratosphere with the help of diabatic heating (Vogel et al., 2019)) complicates atmospheric transport simulations (Willetts et al., 2016).' This sentence sounds odd. I propose to write something like that: 'Monsoon convection transports the boundary layer air to the upper troposphere and to the lower stratosphere, subsequently air parcels are slowly uplifted by diabatic heating to higher altitudes.' What do 'complicates atmospheric transport' mean? Is that related to uncertainties in the representation of convection in atmospheric transport simulations? Please clarify this statement.

We have rephrased the statement and the text is accordingly modified. Also, we emphasized the likelihood of high model uncertainties during the monsoon period due to the possibly inadequate representation of convection in models.

L93-98: "Monsoon convection transports the boundary layer air into the upper troposphere; subsequently air parcels are slowly uplifted by diabatic heating to higher altitudes (Vogel et al., 2019). Representation of these processes in the model is highly complicated, which often leads to transport uncertainties (Willetts et al., 2016). Note that the Asian summer monsoon anticyclone active during the Indian summer monsoon period play a key role in uplifting tracer gases to the upper troposphere and lower stratosphere in most of the cases (Park et al., 2007)."

P4/L127: 'synoptic event' I propose to be here more specific. -> 'the cyclonic storm Ockhi'

**Done.**

In general the western Pacific typhoon season (tropical cyclones) peaks from July to October, therefore I am wondering that the impact of cyclones during July 2017 is not mentioned here. It is discussed later within the paper, but I think it could be mentioned here as well.

We agree and included the following sentence:

**L133-134: "July is also characterized with the presence of strong low-pressure system activity over Bay of Bengal which results in large rainfall availability over central India."**

P6/L213: 'Four global inverse modelling products - CarbonTracker, CarboScope, LSCE v18r3 and LSCE FT18r1- available during the year 2017 are used for our analysis.' I think, it is worth to mention here that none of these models includes ground-based data from the Indian subcontinent. That is first mentioned later within the conclusions. If no ground-based data from the Indian subcontinent are used, it would be helpful to have a comment on the quality of these inverse modelling products, maybe related to other regions around the Indian subcontinent or in the tropics.

We have included this information in the revised manuscript as follows:

L166-169: "It is important to note that none of these products uses ground-based observations from the Indian subcontinent for their optimization, which raises concerns on the reliability of the optimized flux estimations over the region. Suffice it to say that a part of the inter-model differences in predicting the variability can be aroused due to the paucity of  $CO_2$  observations over India."

P10/L360: 'The seasonal variability of CO2 uptake through photosynthesis, release through ecosystem respiration, and the vertical transport is seen while analysing the monthly averaged CO2 concentration profiles over Indian subcontinent (Figs. 2b and 3). Comparatively lower surface CO2 concentrations are found during months with an active biosphere (June to October) than the rest of the period, owing to the more ecosystem productivity over Indian subcontinent in response to the availability of monsoon rainfall.' That is not a central point of the paper, but looking on the Mauna Loa, Hawaii, time series of CO2 (https://gml.noaa.gov/dv/iadv/graph.php? code=MLO&program=ccgg&type=ts) the maxima and minima of monthly averaged CO2 are shifted about  $\sim 4$  weeks later compared to the time series of monthly averaged CO2 concentration at surface over the Indian subcontinent shown in Fig. 2b. Could you make a comment on that?

We added the details and the text is modified as follows:

L418-427: "While comparing the seasonal maximum (May) and minimum (September) of  $CO_2$  concentrations measured at Mauna Loa observatory (MLO) located in Hawaii, Fig. 2b shows a

temporal shift of around one month for exhibiting seasonal maximum (April) and minimum (August)  $CO_2$  concentrations. This temporal shift is attributed to the differential impacts of anthropogenic and terrestrial ecosystem activities on atmospheric concentration as well as the long-distance transfer of atmospheric carbon dioxide to the remote location. Note that MLO observations are generally representative of global mean  $CO_2$  due to the minimal influence of terrestrial ecosystems and anthropogenic activities at the remote location. The seasonal variation of monthly averaged  $CO_2$  seen over the Indian subcontinent is mostly dominant by terrestrial carbon fluxes, i.e., net ecosystem exchange (NEE) as seen from the VPRM simulations (see Supplementary Fig. S1)."

P11/L402: 'Strong mixing and vertical transport associated with the low-pressure systems are visible from these CO2 concentration figures.' Please explain this in more detail. Mark or describe the position of the low-pressure systems in Fig. 5 and 6. What is the role of Asian monsoon anticyclone in vertical (horizontal) CO2 distribution during July? In general, tracer distributions of troposperic source gases in the upper troposphere and lower stratosphere during Asian summer monsoon season depends strongly on the location of the Asian monsoon anticyclone, however not sure what that implies for CO2.

We have marked the positions of low-pressure systems in Figs. 5 and 6 to enhance the clarity. A more detailed discussion of Figs. 5 and 6 is added to the manuscript. The text is modified as follows:

L482-490: "The influence of the synoptic systems on  $CO_2$  concentration can be found during July (Fig. 5a, 80° E to 95° E) and November (Fig. 6b, 75° E to 90° E) with regions exhibiting weak gradients due to strong vertical mixing. As discussed in Sect. 3.1, the presence of enhanced biospheric activity during July reduces the  $CO_2$  concentration in the lower troposphere. Also, the strong vertical and horizontal mixing due to the monsoon circulation dilutes the  $CO_2$ concentration in the atmosphere during July compared to November. The presence of active convection during July transports these less-concentrated air into the upper part of the troposphere, and thereby dilutes the  $CO_2$  concentration values in the upper troposphere as well. This intense convective activity is less pronounced during November except for the events of synoptic storms like Ockhi described below."

L492-496: "Another remarkable feature is the presence of a band of higher representation error along the foothills of Himalayas. In addition to the complex terrain, the region over the Ganges basin is also characterized with increased anthropogenic activity, which contributes to larger representation error surrounding this region. We can also find individual cells with high representation error associated with point emission sources such as cities, mining sites, and coal-based power plants at different parts of the domain."

L93-98: "Monsoon convection transports the boundary layer air into the upper troposphere; subsequently air parcels are slowly uplifted by diabatic heating to higher altitudes (Vogel et al.,

2019). Representation of these processes in the model is highly complicated, which often leads to transport uncertainties (Willetts et al., 2016). Note that the Asian summer monsoon anticyclone active during the Indian summer monsoon period play a key role in uplifting tracer gases to the upper troposphere and lower stratosphere in most of the cases (Park et al., 2007)."

P11/L403: 'Compared to July, we find higher representation error in November owing to the wintertime transport with decreased vertical mixing and less biospheric uptake.' To which Figure do this sentence refer? -> Fig. 7?

This sentence refers to Fig.4. The text is modified as follows to enhance the clarity.

L490-492: "Compared to July, we find higher representation error in November owing to the wintertime transport with decreased vertical mixing along with heterogeneous biospheric uptake (see Fig. 4)"

P13/L466: 'Though the effect of LLJ and TEJ is visible throughout July (Fig. 5b), strong convective activity associated with the low-pressure systems is visible during July 10-18 (Fig. 5a). Please be here a bit more specific and explain what is shown in Fig. 5b. The statement 'is visible' is very general.

The text is modified as follows:

L572-575: "Due to the effect of LLJ and TEJ, we can find a well-mixed region throughout the longitude cross section located at around 850 mb (LLJ) and 200 mb (TEJ) for the entire month of July (Fig. 5b). Additionally, we see a strong convective activity associated with the low-pressure systems during July 10-18 (Fig. 5a)."

P16/L586: 'This indicates that the employed models need to be critically improved in terms of capturing mesoscale phenomena and fine-scale flux variability in order to maximize the potential of deducing the information obtained from these high precision measurements, thereby improving the estimation of surface fluxes.' What about uncertainties in convection?

The text is modified as follows:

L775-778: "Our findings indicate that the models need to be critically improved to capture mesoscale variations associated with horizontal and vertical transport and fine-scale flux variability to maximize the potential of highly precise and accurate measurements."

P31/Fig. 5/6: Please add the used latitude range in the figure captions. Further, I am missing a more detailed discussion of Fig. 5 and 6 e.g. reasons for different CO2 values in July and November. The vertical position of the CO2 maxima is on higher latitudes in November compared to July. Can you comment on that?

We have modified the figure captions as given below. Also included a more detailed discussion of Figs. 5 and 6. Please see the previous comment.

L1168-1172: "Figure 5: Latitudinal averaged (5° N to 40° N) vertical cross-section of  $CO_2$  concentration from WRF-GHG simulations during July 2017. a) During days July 10-18. There are a number of low-pressure systems present over the monsoon-trough region during this period b) During days July 20-28. Low-pressure systems are not so pronounced during this time. The effect of LLJ and TEJ is found throughout the whole month."

L1175-1178: "Figure 6: Latitudinal averaged (5° N to 40° N) vertical cross-section of CO2 concentration from WRF-GHG simulations during November 2017. a) During days November 11-20. There are no low-pressure systems present during this period. b) During days November 21-30. This period is characterized with low-pressure systems in Bay of Bengal and Arabian sea, one of which intensified further and formed into the severe cyclone Ockhi."

**Technical Issues:**

P2/L52: GHG  $\rightarrow$  greenhouse gas (GHG)

Done.

P3/L111: synoptic events  $\rightarrow$  synoptic events (e.g. tropical cyclones)

Done.

P11/L402: gradients..  $\rightarrow$  gradients.

Done.

P12/L437: 6ppm → 6 ppm

Done.

P35/Fig.10: x-axis title: 'CO2(ppm)' -> 'CO2 (ppm)'

Done.

P37/Tab.1: odd line breaking in 'Versio-n'

Done.

P38/Tab.2: odd line breaking in 'CT2019—B'; 'Cabron Tracker' → 'Carbon Tracker'

Done.

**Reviewer 2**

The publication contains a number of interesting and valuable aspects, but it is missing coherence and a clear line of thought. The overall goal of the publication is rather vague, the individual elements are only loosely connected, and some of the analyses need to be better motivated and explained and more thoroughly analyzed. I therefore cannot recommend publication at this stage but suggest major revisions.

Thank you for appreciating the significance of our work. We have considered the comments, worked thoroughly on improving the clarity of the manuscript, and revised it carefully to overcome the shortcomings mentioned above. The changes made to the manuscript are listed.

**Main issues:**

• The overall aim of the paper is not sufficiently clear. On the one hand, the authors emphasize the need of applying high-resolution models (with resolutions of 10 km x 10 km or better), but on the other hand they present a method, how unresolved spatial variability can be accounted for in large-scale models to improve CO2 source/sink estimates.

In my view the paper would gain a lot, if it would much more clearly focus on global coarse-resolution model systems and on how problems of not resolving the small-scale CO2 variability in these models can be mitigated. Although this goal is nicely formulated at the end of page 3, this focus is lost in many of the other sections and especially in the abstract and the conclusions.

Global data assimilation/inverse modeling systems will continue to play an important role. Since the spatial resolution of these systems is continuously increasing, they are more and more applied to study sub-continental or even national scale fluxes. Current systems (4 of them are presented) are typically operating at coarse resolutions of several degrees (i.e. several 100 km), but resolutions of about  $1^{\circ}x1^{\circ}$  (i.e. about 100 km) are quite likely achievable in the near future, so that the analysis of spatial variability below  $1^{\circ}$  as presented in this study is quite relevant.

Many of the elements of the paper could be preserved, but important parts of the text need to be revised or rewritten to sharpen the focus of the manuscript. It is quite disturbing that the need for high-resolution model systems is emphasized over and over again, while the main essence of the paper is to present a method that allows accounting for small-scale, unresolved variability in coarse global models.

Thank you for giving us a chance to explain better. We do agree that the global inverse modelling systems play an important role in studying continental or sub-continental fluxes at annual or sub-annual scales. However, the global models pose the problem of not resolving the fine-scale variability arising both from mesoscale transport and surface fluxes, which limits the

accurate estimation of regional fluxes through inverse modelling. This study investigates on how large is the unresolved variability and its impact on national flux estimation at monthly scale, and explores a potential solution to mitigate the impact of the associated error to improve CO2 source and sink estimates. Our findings emphasize that the representation error due to the sub-grid scale variability needs to be taken into account in inverse models to improve the optimal estimates. The principal solution to address this problem is to increase the resolution of the employed model and our study shows a gradual reduction of representation error when increasing the model's spatial resolution (See Sect. 3.2.1). As an alternative option in favour of current computational capacity, we demonstrate the possibility of a simple parameterization scheme to treat the representation error in global models which resulted in a remarkable reduction of uncertainty in flux estimations (See Sect. 3.5). At the same time, we also point out that such a simple parameterization may not correct certain biases arising due to complex atmospheric flows such as monsoon circulations. In those cases, only simulation resolving these key features is likely to lead to any improvement and the applicability of parameterization is limited. Additionally, we also argue that by merely increasing the resolution will not be helpful without the adequate representation of physical variables and underlying processes.

We have modified the manuscript to highlight the above aspects. Abstract and Conclusions are revised significantly to state these more clearly. Major changes are listed below.

L16-17: "This study aims to investigate the importance of representing fine-scale variability of atmospheric  $CO_2$  in models for the optimal use of observations through inverse modelling."

L34-36: "Efficacy of a simple parameterization scheme to capture the unresolved variability in the coarse models is further illustrated during non-monsoon periods, which reduces the bias in flux estimates from 9.4% to 2.2 %."

L56-57: "These global data assimilation systems play an important role in studying continental or sub-continental fluxes at annual or sub-annual scales."

L260-262: "Most of the current global model simulations are performed at coarse resolutions of several degrees. But with the recent advancement in computational capacity and numerical techniques, a horizontal resolution of to  $1^{\circ} \times 1^{\circ}$  is quite likely achievable for the global model in the near future."

L273-276: "In order to assess the dependence of representation error on horizontal resolution of the employed model, we have computed representation error for multiple resolutions ranging from  $0.5^{\circ} \times 0.5^{\circ}$  to  $4^{\circ} \times 4^{\circ}$ , in addition to  $1^{\circ} \times 1^{\circ}$ , which would encompass the resolutions of both present and near future global inverse modelling systems."

L547-556: "To assess the dependence of representation error on possible horizontal resolutions of the global models, we have further derived the representation errors for different spatial

resolutions of 2°, 3° and 4°. As expected, we see reductions in representation errors for both surface and column averaged CO2 with increasing horizontal resolution of the model (See supplementary Figs. S5 and S6). During July, the median surface representation error reduced from 1.6 ppm (2.5 ppm) to 0.7 ppm (1 ppm) during daytime (nighttime) while increasing horizontal resolution from 4° to 0.5°. This increment in spatial resolution has also resulted in similar error reductions in November during which, the median of surface representation error shows a reduction from 7 ppm (8 ppm) to 1.5 ppm (2 ppm) during daytime (nighttime). In the case of column averaged values, the median representation error decreased from 1.5 ppm (1.4 ppm) to 0.5 ppm (0.4 ppm) during July daytime (nighttime) and from 3.5 ppm (3.8 ppm) to 0.8 ppm (0.7 ppm) during November daytime (nighttime)."

L589-594: "This high representation in the upper troposphere during July is troublesome while utilizing satellite observations, however the availability of satellite observations is also expected to be limited due to more cloud cover during July than rest of the period. This implies that the monsoon circulations, along with extensive convective activities, can dominate the surface flux heterogeneity in producing sub-grid variability, thereby causing significant biases in the flux estimations when transport features are not adequately resolved."

L762-773: "Increasing the spatial and temporal resolutions of the transport models can generally capture the mesoscale features and associated  $CO_2$  gradients, thereby reducing the representation error. This is also reasserted by our estimations of representation error over India for multiple resolutions ranging from 4° to 0.5°. Increasing the model's resolution to  $0.5^{\circ} \times 0.5^{\circ}$  has shown an improvement in capturing variability with representation error reduction of 22% and 47% for summer time and winter time respectively. By showing the existence of considerable magnitude of unresolved variability in 0.5° sub-grid scale with a similar spatial pattern of error as of 1° spatial resolution, we demonstrate the need for a much finer resolution than 0.5° for representing the atmospheric  $CO_2$  variability over India. However, by merely increasing the resolution without having a realistic representation of terrain heterogeneity and flux (both natural and anthropogenic) variability would not be beneficial. Note that the uncertainties in the high-resolution fluxes can worsen the model's skills, whose effect would not be more pronounced at coarser resolutions due to the diffusive nature of fluxes, as seen in Agusti-Panareda et al. (2019)."

L775-778: "Our findings indicate that the models need to be critically improved to capture mesoscale variations associated with horizontal and vertical transport and fine-scale flux variability to maximize the potential of highly precise and accurate measurements."

• The setup of the OSSE described in Section 2.4.1 is not clear at all, and therefore it is impossible to interpret the results. In particular, it is unclear how the simulated observations y\_OSSE were generated. Why are y\_sim and y\_OSSE different, if the same transport model was used to generate them? How exactly was the representation error

accounted for? None of the equations in this section contain a representation error. Did the error change with time or was it set to a monthly mean value? Was the systematic component accounted for or were all errors treated as random? Were temporal correlations in the representation error considered? A careful setup of an OSSE is critical. Based on the information provided, it is not possible to judge whether this was the case (for further specific questions see also my minor comments below).

We have made additional statements and included an equation (Eq. (8)) in the methodology section (Sect. 2.4) to explain the OSSE setup better. Major changes made are listed below.

L322-324: "This means that the difference between observations ( $y_{OSSE}$ ) and simulations ( $y_{sim}$ ) of atmospheric CO2 is the representation error in ppm."

L324-326: "Additionally, as a first-order simplification for the inversion, we assume that the footprints of each observation site span for a radius of 200 km around the site (i.e. the area within a circle of 200 km radius around the site) based ..."

L356-359: "Here we have used WRF-GHG simulations as  $\mathbf{y}_{OSSE}$ .  $\mathbf{y}_{sim}$  is obtained as the sum of  $\mathbf{y}_{OSSE}$  and a realization of the hourly representation error ( $\delta_{CO_2}$ ) at the measurement site as follows:

 $\boldsymbol{y}_{sim} = \boldsymbol{y}_{OSSE} + \boldsymbol{\delta}_{CO_2}$

where  $\boldsymbol{\delta}_{CO_2}$  is calculated using Eq. (1)."

L360-362: "Any temporal correlations in the representation error are not considered for this experiment. We have performed the inversion separately for daytime and nighttime values to identify the impact of diurnal variations of representation errors on flux uncertainty."

• The individual parts of the publication are not sufficiently well connected. For example, the analysis of the differences between the global models in Section 3.1 is interesting by itself, but it is not explained why this is of interest in the context of the overall goal of the paper. Similarly, the discussion of the influence of convective periods in July and of a cyclone in November on the vertical distribution of CO2 and of the representation error is quite interesting, but again there is little discussion of how this relates to the overall scope of the paper.

Global model simulations are compared to assess the variations in monthly averaged  $CO_2$  concentration. Both time-series and vertical profiles show considerable inter-model differences. As discussed in the manuscript, these differences (even seen at the monthly annual scales) indicate significant uncertainties in flux estimations over India. A part of this discrepancy can come from the coarse resolution global model's inability to represent transport processes and

(8)

ecosystem activities that operate at fine scales. This is discussed in the paper ("A part of this discrepancy can come from the coarse resolution global model's inability to represent transport processes like convection and vertical mixing, strength and distribution of anthropogenic sources and ecosystem activities that operate at fine scales."). Also, as a response to the Reviewer 1, we stated the magnitude of the seasonal variability which is comparable to the derived representation error. We revised the manuscript to better highlight the context of our inter-model discrepancies to the rest of the analysis.

We further argue that the variability associated with convection as seen in July is generally difficult to be captured by the coarse models. This is highlighted in Sect. 3.2 by showing the unresolved variability in global models by choosing convective episodes. We agree to add more discussion on these aspects and the text is modified accordingly.

Major changes are listed below.

L21-23: "The extent of monthly averaged surface representation error reaches up to  $\sim 10$  ppm depending on season, which is even comparable to the magnitude of seasonal variability or concentration enhancement due to hotspot emissions."

L161-162: "We have used optimized products at global scales to examine the differences in representation of  $CO_2$  variability by global models over the Indian subcontinent at monthly and annual scales."

*L436-437: "The extent of this unresolved variability in global models is further explored in Sect. 3.2."*

L450-452: "The representation error at  $1^{\circ} \times 1^{\circ}$  spatial scale reaches values ranging from 2.5 ppm to 10 ppm, which are comparable to the magnitude of variability at hotspot emission regions or the seasonal variability CO2 over the region (see Fig. 2b)."

L470-479: "As seen in Fig. 4, we use monsoon (July) and post-monsoon (November) periods for our analysis to identify the seasonal changes in the sub-grid variability over India. The derived seasonal differences in structural patterns of the sub-grid variability facilitate to 1) quantify what would be transport errors associated with incorporating seasonally varying observations into atmospheric models 2) determine what drives the seasonality in sub-grid variability and ultimately 3) design the possible parameterization of representation error with a seasonal component in the inverse modelling framework as well as identify periods or seasons where the use of this parameterization would be valid to improve our estimations of  $CO_2$  fluxes. Further, the seasonal spatial variability analysis of column averages can provide useful information for the satellite community to gap-fill the satellite soundings over India when large data gaps and low sounding precision on daily or monthly time scales are present, which is especially the case for monsoon periods in India." L482-490: "The influence of the synoptic systems on  $CO_2$  concentration can be found during July (Fig. 5a, 80° E to 95° E) and November (Fig. 6b, 75° E to 90° E) with regions exhibiting weak gradients due to strong vertical mixing. As discussed in Sect. 3.1, the presence of enhanced biospheric activity during July reduces the  $CO_2$  concentration in the lower troposphere. Also, the strong vertical and horizontal mixing due to the monsoon circulation dilutes the  $CO_2$ concentration in the atmosphere during July compared to November. The presence of active convection during July transports these less-concentrated air into the upper part of the troposphere, and thereby dilutes the  $CO_2$  concentration values in the upper troposphere as well. This intense convective activity is less pronounced during November except for the events of synoptic storms like Ockhi described below."

L492-496: "Another remarkable feature is the presence of a band of higher representation error along the foothills of Himalayas. In addition to the complex terrain, the region over the Ganges basin is also characterized with increased anthropogenic activity, which contributes to larger representation error surrounding this region. We can also find individual cells with high representation error associated with point emission sources such as cities, mining sites, and coal-based power plants at different parts of the domain."

L723-725: "For instance, we find that the unresolved variations (representation error) of the global models with a spatial resolution of  $1^{\circ} \times 1^{\circ}$  can be  $\sim 3$  ppm on average for the surface CO2 that is even larger than the currently reported inter-global model differences."

*L589-590: "This high representation in the upper troposphere during July is troublesome while utilizing satellite observations..."*

• The analysis of sub-grid scale variability in total column XCO2 as observed by satellites needs to be better motivated. Subgrid-scale variability is an obvious problem when using surface in-situ measurements in a coarse model system, but it is much less obvious for satellite observations. Different from surface in-situ observations, satellite observations (from an imaging satellite) could be averaged over a whole model grid cell, which would alleviate the problem of not resolving sub-grid scale variability. I therefore disagree with the statements on lines 422 to 424.

We disagree with the statement that the impact of subgrid variability is much less obvious for inversions using satellite  $CO_2$  retrievals. Footprints of satellites used for regional flux estimates are much smaller than global model grid cells and the satellite retrievals are only used when clear sky conditions are met. Note that even a small representation error in the column averaged values can lead to large uncertainty in surface flux estimations.

Also note that we report the systematic model transport error in representing  $CO_2$  variability in coarse models. Though random measurement errors will be reduced on averaging, the model errors, particularly systematic errors, remain in the optimization and continue to bias the

estimates. Hence the transport models, in this context, need to be highly accurate in capturing the variability as reported by the observations.

To make it clearer, we have rephrased and added statements in the revised manuscript as follows:

L304-312: "Since satellite footprints are smaller (~  $2 - 20 \text{ km}^2$ ) than the current model grid size (> 100 km), using these measurements for optimization via inverse modelling introduces spatial representation errors and associated uncertainties in the inferred fluxes. Note that the spatial biases of a few tenths of a ppm in column-averaged CO2 can potentially alter even the annual sub-continental fluxes in the range of tenths of a gigaton of carbon fluxes (Chevallier et al, 2007, Miller et al., 2007 and Chevallier et al., 2010). To quantify these systematic transport errors when representing satellite measurements in inverse models, we calculate the spatial representation errors for XCO2 that coarse inverse modelling would suffer from using highly precise and accurate satellite measurements. The impact of these quantified errors on retrieved fluxes is further explored in Sect. 2.4.2."

L516-519: "Note that the representation error reported here is different from satellite measurement errors (e.g. spectroscopic retrieval error or sampling biases) and is systematic in nature. The estimated column representation error is thus capable of causing significant biases in the satellite inferred  $CO_2$  fluxes over India. This result also reasserts the need for taking account of the sub-grid variability  $CO_2$  in models when utilizing satellite observations."

L443-444: "The representation errors for  $XCO_2$  are also calculated separately to assess the ability of current-generation models in utilizing the satellite measurements over India."

• The analysis of the factors influencing the representation errors is too limited and not sufficiently systematic. As shown in different parts of the paper, the errors vary with time (e.g. day vs. night), which meteorology (higher during convective periods), and depend on topography and surface flux variability. It would be useful to analyze the importance of these factors more systematically in a single section. An attempt is made in Section 3.3, but only focusing on the importance of topography. Another attempt is made in Section 3.5, this time using a multivariate model and using total column observations. It is very hard to understand these choices. There should be one single section applying a multivariate model to representation errors in both near-surface CO2 and in total column CO2. Furthermore, it should be analyzed separately how much these factors influence the systematic part of the representation errors and how much they contribute to the random part.

We have revised the manuscript and included additional details explaining the influence of flux variability (both anthropogenic and natural) on the representation error. We have focused on estimating the impact of these variables on the systematic part (on a monthly scale) of the representation error by minimizing the uncorrelated errors (i.e. the monthly averaged representation errors are used for the analysis). Further attempt is done to remove any remaining

part of the random errors (uncorrelated errors) in  $\sigma_{CO_2}$  by calculating representation error using monthly averaged CO2 concentrations (denoted as  $\sigma_{\overline{CO_2}(mon)}$ ). Though we haven't focused on the purely random part of the representation error here, supplementary Table S2 is provided to summarize separately how much the explanatory variables influence the systematic part of the representation errors and how much they contribute to the random part.

L285-289: "We consider that  $\sigma_{CO_2}$  is not purely systematic, and may contain small fraction of random components though random variability can be largely minimized through temporal averaging as done in Eq. (2). Further we have calculated the representation error using monthly averaged CO2 concentrations (see Eq. (3)). The estimated representation error ( $\sigma_{\overline{CO_2}(mon)}$ ) is particularly useful in inversion when deriving averaged fluxes over a long time period, e.g. monthly fluxes."

L596-600: "Here we explore the factors influencing the size and patterns of the representation error in coarse models. For this, statistical relationships between representation error and possible explanatory variables are examined for both surface and column-averaged  $CO_2$ . Identifying these factors influencing representation errors and quantifying their local effects facilitate us to further investigate on how these biases in retrieved fluxes can be minimized in global models (see Sect. 3.5)."

L623-636: "Further, we estimate the statistical relationship between the surface flux heterogeneity and representation error. We find that the correlation between heterogeneity in biospheric surface flux (as derived from the standard deviation of VPRM-derived NEE fluxes, denoted as  $\sigma_{bio}$ ) and representation errors depends on time of the day and season. During daytime in July, the dependence of  $\sigma_{bio}$  on representation error ( $\sigma_{CO_2}$ ) of surface and column  $CO_2$  is found to be 19 % and 42% respectively, while the correlation is negligible during nighttime (4 % for the surface  $CO_2$  variability and 11 % for the column variability, which is consistent with the findings of Pillai et al., (2010). The diurnal difference of the dependence of  $\sigma_{bio}$  on representation error can be explained by the increased magnitude and spatial variability of daytime biospheric fluxes in growing season (primarily due to photosynthesis activities) compared to nighttime fluxes. Moreover, poor vertical mixing under the stable nocturnal atmospheric conditions with more advection and drainage flow reduces the influence of surface fluxes on spatial variability of mixing ratios. In general, the local influence of biosphere fluxes on spatial variability of  $CO_2$  for surface and column  $CO_2$  is found to be less ( $\sigma_{CO_2}$  and  $\sigma_{bio}$  are correlated negligibly) in November due to the effect of cyclonic storm. During daytime in noncyclonic period of November, a considerable correlation is found between  $\sigma_{CO_2}$  and  $\sigma_{bio}$  (54 % for surface and 37 % for column)."

L637-644: "We find less influence of seasonality on relationship between anthropogenic surface flux heterogeneity (as derived from the standard deviation of EDGAR fluxes, denoted as  $\sigma_{ant}$ )

and representation errors. The dependence of the anthropogenic flux heterogeneity on representation is found to be not considerable except for July daytime (16–19%), showing the dominance of meteorological and biospheric flux variability on causing representation error. Similar to above analysis with  $\sigma_{bio}$ , the combined effect of atmospheric stability and flux heterogeneity can explain the diurnal differences of the relationship between  $\sigma_{ant}$  and  $\sigma_{CO_2}$ . While the representation is explained by  $\sigma_{ant}$  (7–19% of the total surface variability and 8–16% % of the surface variability) during daytime, this relationship is absent during nighttime (see Supplementary Table S1)."

L645-650: "In case of variability of monthly averages, we see that  $\sigma_{\overline{cO_2}(mon)}$  is well explained by  $\sigma_{bio}$  during daytime (see Supplementary Table S2), as expected. A similar strong correlation can be seen between  $\sigma_{\overline{cO_2}(mon)}$  and  $\sigma_{bio}$  (65–67 %) during nighttime for surface variability of  $CO_2$ , while there exists only negligible dependence of local fluxes on nocturnal column  $CO_2$ variability. This shows the decoupling of the mixing ratios in other part of column from surface during night due to less vertical mixing, combined with more drainage flow in the nocturnal boundary layer, which reduces the effect of surface flux variability on column  $CO_2$  variability."

L688-691: "Similarly, we have modelled the surface representation error using the linear model with three explanatory variables as mentioned above and found that the proposed model could capture the derived surface representation error well with a bias less than 1 ppm in most of the regions (see Supplementary Fig. S10 and Supplementary Table S1 and S2)."

**Minor points:**

• Abstract, Line 21: The reader doesn't know at this point which coarse models are meant, and therefore one cannot write "THE coarse models". The definite article "the" is wrongly used at many other places in the manuscript. I trust that the manuscript will be checked for grammar before a possible publication.

Thank you for the suggestion. We have taken note of that. The manuscript is modified accordingly.

L17-19: "The unresolved variability of atmospheric  $CO_2$  in coarse models is quantified by using WRF-Chem simulations at a spatial resolution of 10 km  $\times$  10 km."

• Abstract, line 22: Typical/average/median values are much more relevant than extreme values.

The text is modified with median values.

L19-21: "We show that the representation error due to unresolved variability in coarse model at a horizontal resolution of one degree ( $\sim 100 \text{ km}$ ) reaches a median value of 3 ppm and 1.3 ppm

for the surface and column  $CO_2$  respectively, which are considerably larger than the measurement errors."

• Abstract, line 23: What is a "sampling error"? Here and at many other places this should be replaced by "measurement error".

Done.

• Page 2, lines 66-70: Both variations in orography (affecting the flow) and in land use (affecting the fluxes) are important. These two different factors should be more clearly distinguished and described.

The text is modified as follows incorporating the reviewer's suggestion.

L68-70: "Variations in topography influences the transport of the tracers and when the small-scale orographic details are not adequately represented..."

L73-75: "Further, the variations in land use patterns for neighbouring regions can cause considerable variability in the  $CO_2$  surface fluxes. Thus, a proper representation of land use patterns in model is also important in terms of simulating  $CO_2$  variability."

• P3, L80: replace "from the last decade" by "during the last decade"

Done.

• P3, L83: replace "these coarse models on representing" by "coarse global models in representing"

Done.

• P3, L97: Please explain in which way the dry and wet seasons affect the cropping patterns. Is cropping enhanced during the wet or during the dry season? Or does this depend on the type of crop?

Additional information is added to the manuscript.

L105-108: "Wet season crops (Kharif crops during June to November, e.g. Rice, Millets, and Maize) in India mainly depend on monsoon rain and dry season crops (Rabi crops during November to April, e.g. Wheat, Barley, and Mustard) are less water dependent and mainly rely on the water irrigation."

• P3, L101: replace "The study" by "This study".

Done.

• P4, L121: replace "of the high-resolution" by "of high-resolution"

**Done.**

• P4, L122: replace "is characterized" by "was characterized"

**Done.**

• P4, L134: Please reformulate the sentence. Estimating an assessment doesn't make sense.

The text is modified as follows:

L142-144: "Through the global model comparisons and spatial variability analysis, we highlight potential difficulties for estimating  $CO_2$  budget over India..."

• P4, L141: replace "from the inverse" by "from inverse" (again a wrong use of the definite article) and replace "estimates" by "estimate"

**Done.**

• Section 2: Consistent with the emphasis on global models, the global model systems should be described before the WRF-Chem model system.

Manuscript is revised accordingly.

• Section 2: The global models need to be described in more detail. E.g. which observations were assimilated is not always described. Furthermore, what was the driving meteorology in these offline transport models? Could this explain the large differences? Or is it the fact that the models use different convection and PBL turbulence parameterizations? It would be good to summarize the main features of the models (resolution, driving meteorology, parameterizations, emission inputs, biospheric flux models, assimilated observations) in a table.

We have incorporated the suggestions from the reviewer. Table. 1 in the manuscript provides details of the global models used in this study.

• P5, L154: Why should entropy be conserved? As far as I know, the Skamarock report doesn't mention any conservation of entropy.

Yes, we agree. The text is modified as follows:

L195-196: "For simulating the atmospheric transport, the model uses fully compressible Eulerian non-hydrostatic equations on Arakawa C- staggered grid, conserving mass, momentum and scalars (Skamarock et al., 2008)."

• P5, L168: replace "is also established" by "was established"

Done.

• P5, L177: How is SYNMAP mapped onto the 0.1° grid? Was only the dominant land cover type used, or is a tile approach implemented in WRF-GHG, i.e. an approach accounting for all different land cover types within the 0.1° grid cell?

We have used a tile approach to map the SYNMAP on to WRF-GHG grid. The text is modified as follows for more clarity.

L219-221: "VPRM uses the SYNMAP vegetation classification (using tile approach) (Jung et al., 2006) as well as EVI and LSWI from MODIS surface reflectance data at a resolution of 1 km and 8 days."

• P6, L207: ".. a different simulation strategy .. ". Different from what?

The text is modified as follows:

L248-249: "We have utilized a simulation strategy to update the initial meteorological conditions for taking advantage of assimilated meteorological fields from ECMWF."

• P7, L243: I disagree that sub-grid scale variability is "fully resolved" by the high-resolution model. Actually, a model at 10 km resolution is not at all sufficient to resolve mesoscale flows in mountainous terrain. It needs to be explained that the simplifying assumption is made that the high-resolution model captures a major part of sub-grid scale variability, but that the true variability is likely larger, since even a model at 10 km resolution cannot resolve all variability.

We agree. The text is modified as follows:

L258-260: "It is assumed that the high-resolution simulation captures the majority of the subgrid scale variability even though it cannot be expected to resolve all observed variability."

• P7, L244: The choice of a resolution of 1°x1° to study sub-grid scale variability is poorly motivated. Why not 2°? Why not 0.5°? Actually, the paper would gain a lot if it would study the variability at multiple resolutions between 0.5° and 4°, which would encompass the resolution of both present and (near) future global inverse modelling systems.

We have also conducted some additional analysis for multiple resolutions between  $0.5^{\circ}$  and  $4^{\circ}$ . Accordingly, we have revised the manuscript as follows: L260-262: "Most of the current global model simulations are performed at coarse resolutions of several degrees. But with the recent advancement in computational capacity and numerical techniques, a horizontal resolution of  $1^{\circ} \times 1^{\circ}$  is quite likely achievable for the global model in the near future."

L273-276: "In order to assess the dependence of representation error on horizontal resolution of the employed model, we have computed representation error for multiple resolutions ranging from  $0.5^{\circ} \times 0.5^{\circ}$  to  $4^{\circ} \times 4^{\circ}$ , in addition to  $1^{\circ} \times 1^{\circ}$ , which would encompass the resolutions of both present and near future global inverse modelling systems."

L547-556: "To assess the dependence of representation error on possible horizontal resolutions of the global models, we have further derived the representation errors for different spatial resolutions of 2°, 3° and 4°. As expected, we see reductions in representation errors for both surface and column averaged  $CO_2$  with increasing horizontal resolution of the model (See supplementary Figs. S4 and S5). During July, the median surface representation error reduced from 1.6 ppm (2.5 ppm) to 0.7 ppm (1 ppm) during daytime (nighttime) while increasing horizontal resolution from 4° to 0.5°. This increment in spatial resolution has also resulted in similar error reductions in November during which, the median of surface representation error shows a reduction from 7 ppm (8 ppm) to 1.5 ppm (2 ppm) during daytime (nighttime). In the case of column averaged values, the median representation error decreased from 1.5 ppm (1.4 ppm) to 0.5 ppm (0.4 ppm) during July daytime (nighttime) and from 3.5 ppm (3.8 ppm) to 0.8 ppm (0.7 ppm) during November daytime (nighttime)."

• P7, L246: The equation describes the standard deviation at any given instance in time. Later in the paper, a distinction is made between random and systematic variations. How these separate components are computed needs to be explained in this section, too.

We have made additional details and revised the manuscript accordingly.

L262-269: "For estimating the representation error in a coarse model with a typical spatial resolution of  $1^{\circ} \times 1^{\circ}$ , we have calculated the standard deviation of  $CO_2$  dry air mole fraction simulated by the WRF-GHG model within the coarse grid boxes of  $1^{\circ} \times 1^{\circ}$  as follows:

$$\sigma_{CO_{2(tot)}} = \sqrt{\frac{1}{n-1} \sum_{j=1}^{n} (m_j - \overline{m})^2}$$

$$where \ \overline{m} = \frac{1}{n} \sum_{j=1}^{n} m_j$$

$$(1)$$

*n* is the number of 0.1° boxes inside the coarser grid cell of  $1^{\circ} \times 1^{\circ}$ ; *m* is the CO2 dry air mole fraction corresponding to 0.1° boxes; and  $\overline{m}$  is the average within the coarser grid cell."

*L281-295: "To disentangle the correlated term from the total representation error* ( $\sigma_{CO_{2(tot)}}$ ), the monthly averages of representation errors ( $\sigma_{CO_{2}}$ ) are taken. The effect of random errors can be minimized when averaging over long time periods.

$$\sigma_{CO_2} = \frac{1}{T} \sum_{t=1}^{T} \sigma_{CO_{2(tot)}} \tag{2}$$

where T is the total number of simulations in a month during daytime or nighttime. We consider that  $\sigma_{CO_2}$  is not purely systematic, and may contain small fraction of random components though random variability can be largely minimized through temporal averaging as done in Eq. (2). Further we have calculated the representation error using monthly averaged  $CO_2$ concentrations (see Eq. (3)). The estimated representation error  $(\sigma_{\overline{CO_2}(mon)})$  is particularly useful in inversion when deriving averaged fluxes over a long time period, e.g. monthly fluxes.

$$(\sigma_{\overline{CO_2}(mon)} = \sqrt{\frac{1}{n-1} \sum_{j=1}^n (M_j - \overline{M})^2}$$
(3)

where  $\overline{M} = \frac{1}{n} \sum_{j=1}^{n} M_j$

n is the number of 0.1° boxes inside the coarser grid cell of  $1^{\circ} \times 1^{\circ}$ ; M is the monthly averaged  $CO_2$  dry air mole fraction at a 0.1° spatial scale; and  $\overline{M}$  is the corresponding average within the coarser grid cell of 1°. The difference between Eq. (1) and Eq. (3) is that we use monthly averaged  $CO_2$  concentration values in Eq. (3) instead of hourly values as in Eq. (1)."

• P7, L255: It is hard to believe that the center of the second layer is at 200 m. The lowest model levels should be much narrower in order to properly capture the diurnal dynamics of the atmospheric boundary layer.

Here we mentioned the average height from mean sea level for the whole domain. The text is modified as follows for more clarity.

**L277-278: "...model level (mean height is ~200 m from sea level)"**

• P7, L258: I don't understand how the correlated term was deduced. Please explain clearly, ideally by providing a formula. The correlated (systematic) component seems very important to me, and therefore should be introduced properly.

We have provided the formula (Eq. (3) in the manuscript). Also, please see our response above.

• Section 2.4.1: The title of this section should be changed to "Generation of pseudoobservations" or something similar. As mentioned earlier, the setup of the OSSE is not clear at all. The description needs to be improved significantly. The title is modified as "Using pseudo surface measurements". We have improved the description of OSSE by adding additional information and notations for variables. Major changes are as follows:

L322-324: "This means that the difference between observations ( $y_{OSSE}$ ) and simulations ( $y_{sim}$ ) of atmospheric  $CO_2$  is the representation error in ppm."

L324-326: "Additionally, as a first-order simplification for the inversion, we assume that the footprints of each observation site span for a radius of 200 km around the site (i.e. the area within a circle of 200 km radius around the site) based ..."

L356-359: "Here we have used WRF-GHG simulations as  $\mathbf{y}_{OSSE}$ .  $\mathbf{y}_{sim}$  is obtained as the sum of  $\mathbf{y}_{OSSE}$  and a realization of the hourly representation error ( $\delta_{CO_2}$ ) at the measurement site as follows:

 $\boldsymbol{y}_{sim} = \boldsymbol{y}_{OSSE} + \boldsymbol{\delta}_{CO_2}$

(8)

where  $\boldsymbol{\delta}_{CO_2}$  is calculated using Eq. (1)."

L360-362: "Any temporal correlations in the representation error are not considered for this experiment. We have performed the inversion separately for daytime and nighttime values to identify the impact of diurnal variations of representation errors on flux uncertainty."

• P8, L285: I don't understand what is meant by "50-90 percentile". I guess one should either use the 50% percentile or the 90% percentile, but why would one use a 50-90% range? Furthermore, it remains unclear whether a "radius of 200 km" was used (i.e. the area within a circle) or really a site-specific area derived from the mean station footprint.

Yes, we have used 50 percentiles. We edited the text accordingly. Also, the text is modified to enhance the clarity as follows:

L325-326: "...200 km around the site (i.e. the area within a circle of 200 km radius around the site)..."

• P8, L290: Replace "Through our .. approach" by "Through a .. approach (see Eq. 2)"

Done.

• P8, L292: Why do you use all hourly values and not only afternoon values here? The results of the inversion critically depends on the choice of observations, and especially on whether the assumed errors are temporally uncorrelated or not. For hourly data, it is very likely that the (spatial representation) errors are temporally correlated. For the OSSE to

be meaningful, any spatial and temporal correlations of the errors need to be properly accounted for.

We have conducted OSSE for daytime and nighttime separately.

L360-362: "Any temporal correlations in the representation error are not considered for this experiment. We have performed the inversion separately for daytime and nighttime values to identify the impact of diurnal variations of representation errors on flux uncertainty."

• Section 2.4.2: It is not sufficiently clear how the pseudo satellite observations were created. What do you mean by "dense" spatial sampling? At the density of OCO-2? What do you mean by "as frequently as possible"? Every hour of the day? Once a day? These formulations are too vague.

The text is modified as follows:

L374-376: "Pseudo satellite observations in this experiment have a spatial resolution as same as the WRF-GHG simulations and the frequency of observation is daily."

• P9, L327: I don't think that retrievals in the short-wave infrared range are sensitive to molecular (i.e. Rayhleigh) scattering, but of course they are sensitive to molecular absorption (by CO2, H2O, O2, etc.)

The text is modified as follows incorporating the reviewer's suggestion.

L377-378: "In practice, the satellite spectra are altered by molecular absorption by air molecules and the presence of aerosols and clouds, which contribute to the measurement error."

• P9, L333: A cloud fraction threshold of 20% is much too high for satellite XCO2 retrievals. Usually the thresholds are in the low percentage range (e.g. 2% cloud fraction), because uncertainties in photon paths are quickly increasing even when thin cirrus is present.

We have sampled observations using a threshold of 2% cloud fraction; however, this criterion resulted in less than 1% of the (pseudo) observations. We indicated the possible reduction of observations in the revised manuscript as follows:

L385-387: "The selected cloud fraction threshold is likely on the higher side than the one usually used for the satellite retrievals; hence a further reduction of observations can be expected in the actual cases."

• P10, L336: replace "significantly low" by "too low"

Done.

• P10, L344: The performance of the models is not assessed in this section. This would require a comparison against observations.

The text is modified as follows:

L396-397: "We first analyse the level of agreement among current-generation global transport models in simulating  $CO_2$  concentration over Indian subcontinent."

• P10, L346: Although quite plausible, it is only a hypothesis that the models have large common model errors. But here it is stated as a fact.

We have modified the text as follows:

L397-399: "Note that a mere agreement among the coarse models is not sufficient to justify the models' performance over the region due to their plausibly large model errors in common and interdependency in terms of data sources."

• P10, L353: Delete "by different models". This is clear from the context.

Done.

The analysis of the differences between the global models is interesting and would deserve a bit more discussion. Furthermore, in order to better integrate this section into the paper, it would be important to compare these differences with the magnitude of the representation errors due to sub-grid scale variability. The differences between the models are surprisingly large, especially during the monsoon season. How plausible are the strong vertical gradients of 2 – 3 ppm below 700 hPa in the LSCE model in July and August? Wouldn't one expect a well-mixed atmospheric boundary layer during the monsoon season in the afternoons?

We have included the following text to the manuscript.

L723-725: "For instance, we find that the unresolved variations (representation error) of the global models with a spatial resolution of  $1^{\circ} \times 1^{\circ}$  can be ~3 ppm on average for the surface  $CO_2$  that is even larger than the currently reported inter-global model differences."

L450-452: "The representation error at  $1^{\circ} \times 1^{\circ}$  spatial scale reaches values ranging from 2.5 ppm to 10 ppm, which are comparable to the magnitude of variability at hotspot emission regions or the seasonal variability  $CO_2$  over the region (see Fig. 2b)."

L430-432: "The strong vertical gradient in the surface levels as simulated by the LSCE model during monsoon period is less plausible and likely be attributed to the absence of observations over India to constrain the model output."

• P10, L363: The CO2 concentrations are not only lower in Jun – Oct due to the active biosphere over India but due to the biosphere over the whole northern hemisphere.

The text is modified as follows:

L414-416: "Comparatively lower surface  $CO_2$  concentrations are found during months with an active biosphere (June to October) than the rest of the period, owing to the more ecosystem productivity over the northern hemisphere and particularly over Indian subcontinent in response to the availability of monsoon rainfall."

• P11, L370: Replace "the significant" by "significant"

Done.

• P11, L378: None of the current generation global models used in this study has a resolution of 1°x1°. As mentioned earlier, it would be useful to analyze how the representation error changes with resolution rather than just presenting the results for one rather arbitrarily selected resolution.

As mentioned above, we have conducted additional analysis of representation error for different resolutions ranging from  $0.5^{\circ}x0.5^{\circ}$  to  $4^{\circ}x4^{\circ}$ . The results of the analysis are described in Sect. 3.2.1.

L547-556: "To assess the dependence of representation error on possible horizontal resolutions of the global models, we have further derived the representation errors for different spatial resolutions of  $2^{\circ}$ ,  $3^{\circ}$  and  $4^{\circ}$ . As expected, we see reductions in representation errors for both surface and column averaged CO2 with increasing horizontal resolution of the model (See supplementary Figs. S4 and S5). During July, the median surface representation error reduced from 1.6 ppm (2.5 ppm) to 0.7 ppm (1 ppm) during daytime (nighttime) while increasing horizontal resolution from  $4^{\circ}$  to  $0.5^{\circ}$ . This increment in spatial resolution has also resulted in similar error reductions in November during which, the median of surface representation error shows a reduction from 7 ppm (8 ppm) to 1.5 ppm (2 ppm) during daytime (nighttime). In the case of column averaged values, the median representation error decreased from 1.5 ppm (1.4 ppm) to 0.5 ppm (0.4 ppm) during July daytime (nighttime) and from 3.5 ppm (3.8 ppm) to 0.8 ppm (0.7 ppm) during November daytime (nighttime)."

• P11, L391: replace "high values" by "higher values"

Done.

• Figure 4 (and Fig. 7) needs to be discussed more thoroughly. One of the remarkable differences between July and November is the much larger representation errors in November along a band extending almost through the whole model domain. It is not clear to me whether this is band is along the border of the Himalayan or along the Ganges river. There are also individual cells with much higher values compared to their neighboring cells. Is this due to anthropogenic sources (cities, industries), due to topography, or due agriculture?

**Manuscript is revised as follows:**

L492-496: "Another remarkable feature is the presence of a band of higher representation error along the foothills of Himalayas. In addition to the complex terrain, the region over the Ganges basin is also characterized with increased anthropogenic activity, which contributes to larger representation error surrounding this region. We can also find individual cells with high representation error associated with point emission sources such as cities, mining sites, and coal-based power plants at different parts of the domain."

• P11, L402: replace "well mixed vertical gradients" by "weak gradients due to strong vertical mixing". The next sentence could probably be deleted.

Done.

• P11, L404: Why should "less uptake" in November compared to July cause "higher representation errors"? I would expect the opposite.

The text is modified as follows:

L490-492: "Compared to July, we find higher representation error in November owing to the wintertime transport with decreased vertical mixing along with heterogeneous biospheric uptake (see Fig. 4)."

• P12, L413: replace "vertical flows" by "vertical transport"

**Done.**

• P12, L415: replace "significantly large" by "significantly larger"

**Done.**

• P12, L418: Delete "of surface sub-grid variations". This is clear without saying.

Done.

• P12, L431-433: The sentence does not make sense to me. Why should the computation of a correlated representation error reduce the effect of the random errors?

The text is modified as follows:

*L525-528:* "To further reduce the effect of random error that might be introduced by short-term weather phenomena, the representation errors ( $\sigma_{\overline{CO_2}}(mon)$ ) are calculated from the monthly averaged CO2 field and are denoted as a systematic error (Fig. 8)."

• The distinction between random and systematic components in the representation error seems very important to me, since the influence of random errors can be compensated by large numbers (of observations), whereas the systematic component likely leads to systematic biases in the flux estimates. These aspects deserve much more attention in the paper and it should be clearly explained how they are calculated (as mentioned earlier).

We have revised the manuscript and included additional details to improve the description. Please see the response to previous comment.

• P13, L450: Replace "sampling errors" by "measurement errors" (here and throughout the manuscript), and replace "significantly high" by "significantly higher".

Done.

• Figure 8: The figure caption says "Variability .. over India", but it is not clear whether this variability was really computed for India only (using e.g. a country mask) or for the whole model domain including the Himalayas and the ocean parts (which both should be excluded from the analysis).

We have sampled the representation error only from India using country mask. The text is modified as follows for more clarity.

L520-521: "Figure 8 shows the statistical distribution of the representation error ( $\sigma_{CO_2}$ ) sampled over India, during July and November, separated by daytime and nighttime."

• P13, L454: replace "minimal" by "relatively small"

Done.

• Figure 9: Figure titles like "Sur-July" or "ColAvg-July" should be replaced by more explicit titles, e.g. "Surface – July" and "Column average – July".

Done.

• P13, L457: replace "synoptic systems prevailed" by "prevailing synoptic systems"

Done.

• P13, L461: A resolution of 1°x1° should generally be sufficient to represent synoptic events. It is actually not so clear to me why the cyclone so strongly influenced subgrid-scale variability. Is it because there were many individual convective cells, or narrow frontal lines? The strong increase in the median value is really remarkable, which suggests that more than 50% of the area of India were affected by the event.

The subgrid variability is caused here due to the small-scale processes associated with the synoptic systems that cause large representation errors in the simulations. We have modified the text for clarity as follows:

L565-567: "...which is difficult to be handled by current generation global models because of the spatial-scale mismatches between model and small-scale processes associated with synoptic events."

• Section 3.2.3: The differences in the vertical profiles of the representation error between July and November should also be discussed in the context of surface versus total column CO2 observations. The large representation errors in the upper troposphere in July are problematic for satellite observations but not for surface observations. On the other hand, there are probably no satellite observations available in July due to cloud cover.

The text is modified as follows:

L589-591: "This high representation in the upper troposphere during July is troublesome while utilizing satellite observations, however the availability of satellite observations is also expected to be limited due to more cloud cover during July than rest of the period."

• P14, L487: replace "spatial figures" by "spatial maps"

Done.

• P14, L489: It is not only mesoscale circulations which influence the spatial variability over hilly terrain, but also the simple fact that the lowest model layer is at a higher altitude over a mountain than over a valley. Total columns are also affected by the same effect. Actually, I suspect that this effect is more important than mesoscale circulations.

We agree that the incorrect representation of the vertical grid can cause additional model errors. However, this additional error is not a part of our representation error estimates since we have used the same vertical resolution of the high-resolution model (WRF-GHG) for our representation error calculations. Note that the model uses the terrain following vertical coordinates. The text is modified as follows:

L604-607: "At the same time, noteworthy is that there is a plausible additional error term related to the insufficient resolution of vertical grids in models to account for a variety of surface influence on simulations (e.g. mountain vs. valley). This effect of coarse vertical resolution is excluded in our representation error estimates by preserving the vertical grids used for the high-resolution simulations."

• Section 3.4: As mentioned earlier, it is very difficult to interpret the results presented in this section without knowing how exactly the OSSE was set up. Furthermore, without knowing what fraction of the area of India is covered by the footprints of the nine stations, the reported flux uncertainties for whole India of 14.5 to 16.2% in July and 6.3 to 7.5% in November are quite meaningless. What is the typical uncertainty for the individual regions? Why are the uncertainties so high, if the spatial representation errors are primarily random and if there is such a large number of observations constraining the fluxes each month?

Please see our previous response related to OSSE.

• P15, L530: replace "take to the account of" by "account for the"

Done.

• Conclusions section:

L580: As in the abstract, it would be more useful to mention typical (median) values rather than just the extreme values.

The text is modified as follows:

L723-726: "For instance, we find that the unresolved variations (representation error) of the global models with a spatial resolution of  $1^{\circ} \times 1^{\circ}$  can be  $\sim 3$  ppm on average for the surface  $CO_2$  that is even larger than the currently reported inter-global model differences. Similarly, the average representation error estimated for the column-averaged  $CO_2$  is  $\sim 1.3$  ppm."

**References**

- Chevallier, F., Bréon, F. M., and Rayner, P. J.: Contribution of the Orbiting Carbon Observatory to the estimation of CO2 sources and sinks: Theoretical study in a variational data assimilation framework, Journal of Geophysical Research, 112, D09307, doi:10.1029/2006JD007375, 2007.
- Miller, C. E., Crisp, D., DeCola, P. L., Olsen, S. C., Randerson, J. T., Michalak, A. M., Alkhaled, A., Rayner, P., Jacob, D. J., Suntharalingam, P., Jones, D. B. A., Denning, A. S., Nicholls, M. E., Doney, S. C., Pawson, S., Boesch, H., Connor, B. J., Fung, I. Y., O'Brien, D., Salawitch, R. J., Sander, S. P., Sen, B., Tans, P., Toon, G. C., Wennberg, P. O., Wofsy, S. C., Yung, Y. L., Law, R. M.: Precision requirements for space-based XCO2 data, Journal of Geophysical Research, 112, D10314, doi:10.1029/2006JD007659, 2007.
- Park, M., Randel, W. J., Gettelman, A., Massie, S. T., and Jiang, J. H.: Transport above the Asian summer monsoon anticyclone inferred from Aura Microwave Limb Sounder tracers, Journal of Geophysical Research, 112, D16309, https://doi.org/10.1029/2006JD008294, 2007.

**Figures and Tables**

Figure 5: Latitudinal averaged (5° N to 40° N) vertical cross-section of  $CO_2$  concentration from WRF-GHG simulations during July 2017. a) During July 10-18. There are a number of low-pressure systems present over the monsoon-trough region during this period b) During July 20-28. Low-pressure systems are not so pronounced during this time. The effect of LLJ and TEJ is visible throughout the whole month.

---

## Author Comment (AC2)

**Authors' response to Yogesh K. Tiwari**

We thank Yogesh K Tiwari for posting the comment. Also, we appreciate the Editor for giving us an opportunity to address this comment and improve the quality of the discussion thread.

We address the comments as follows (comments are given in regular black font, our responses are given in regular blue font, and the changes in the revised version are given in *blue italic font*).

Thank you

Authors

**Comments**

This is a very interesting and timely paper for the regional carbon cycle and flux inversions. The authors have focused to obtain representative errors in the models leading to significant errors in the source/sink estimations in the inversions. Notably, they have quantified that these REs can be as large as 9 ppm which is well above the observational uncertainty and as comparable to enhancement at any point source region.

Thank you for appreciating the importance of our work.

Unfortunately, the study fails to acknowledge recent other similar studies in the Indian flux inversions. To be specific, Nalini et al., (2019) quantified the potential uncertainty reduction achievable by using data from existing tower-based observations over India. Moreover, they also have proposed 17 new stations spread across various parts of India and put forward this important recommendation to initiate observations to benefit Indian flux inversions especially when satellite constellations fail to capture Indian footprints during the heavy rainy season. A large ensemble of particle trajectories and Bayesian inversions with incremental optimization is done in their OSSE work. It is pity that the authors fail to discuss this paper in their work.

Following the above comment, we considered the reference that is recommended to be cited (Nalini et al., 2019) and made thorough scrutiny of the full article. This brings us to perceive that the work reported in the above article is minimally relevant to our work for the following reasons:

1) Our work addresses the unresolved sub-grid scale variability in the current inverse global models when assimilating observations to retrieve the source-sink distribution of $CO_2$ over India. The recommended article studies how an optimal network of observations can

be achieved over India, which can reduce the uncertainty of surface flux estimation. {completely different objectives}

2) In our manuscript, we report the representation errors that can be expected in global data assimilation/inverse modeling systems, discuss their impact on flux estimations, and propose an approach that can minimize the uncertainty due to unresolved $CO_2$ variability. Considering the article in the recommendation, we do not find results in common to be compared or discussed in our manuscript. {outside of the scope for comparing or discussing the results}

3) As may be evident for addressing different focus and objectives, we follow completely different methodology and modelling approaches compared to the article recommended to be cited. {irrelevancy for citing or comparing methodology or approaches}

We, therefore, disagree with the need for the citation of Nalini et al., (2019) in our manuscript.

Tiwari et al, (2013), Ravi Kumar et al., (2016) also discussed the atmospheric concentration variability over India at seasonal and intra-seasonal scales. The authors also discuss the similar RE variation at seasonal and inter-monthly scale due to changes in meteorology. Therefore the above papers should be mentioned in this context.

Thank you for notifying the published information. We do not find that the mentioned research articles report RE variation or their impact. Nevertheless, we see that Ravi Kumar et al., (2016) could be considered to be cited when mentioning intra-seasonal variability of atmospheric $CO_2$. The citation is included in the revised manuscript.

*L102-103: "... spatio-temporal patterns of the biogenic fluxes (e.g., Gadgil, 2003; Valsala and Maksyutov, 2013, Ravi Kumar et al., 2016)."*

References

Nalini, K., S. Sijikumar, Vinu Valsala, Yogesh K. Tiwari, Radhika Ramachandran, 2019, Designing surface CO2 monitoring network to constrain the Indian land fluxes, Atmospheric Environment 218 2019. 117003, https://doi.org/10.1016/j.atmosenv.2019.117003, 28 Sept., 2019

Ravi Kumar K., Valsala V., Yogesh K. Tiwari, Revadekar J.V., Pillai P., Chakraborty S., Murtugudde R., 2016, Intra-seasonal variability of atmospheric CO2 concentrations over India during summer monsoons, Atmospheric Environment, 142, October 2016, ISSN:1352- 2310, DOI:10.1016/j.atmosenv.2016.07.023, 229-237

Yogesh K. Tiwari, Vinu Valsala, Ramesh K. Vellore, Ravi K. Kunchala, 2013, Effectiveness of surface monitoring stations in capturing regional CO2 emissions over India. Climate Research, Vol. 56: 121–129, 12 March 2013, ISSN 1616- 1572, DOI: 10.3354/cr01149